# ADAPTIVE PRUNING OF PRETRAINED TRANSFORMER VIA DIFFERENTIAL INCLUSIONS

**Yizhuo Ding, Ke Fan, Yikai Wang✉, Xinwei Sun✉, Yanwei Fu**
School of Data Science, Fudan University
{yzding22,kfan21}@m.fudan.edu.cn; yi-kai.wang@outlook.com;
{sunxinwei,yanweifu}@fudan.edu.cn

## ABSTRACT

Large transformers have demonstrated remarkable success, making it necessary to compress these models to reduce inference costs while preserving their performance. Current compression algorithms prune transformers at fixed compression ratios, requiring a unique pruning process for each ratio, which results in high computational costs. In contrast, we propose pruning of pretrained transformers at any desired ratio within a single pruning stage, based on a differential inclusion for a mask parameter. This dynamic can generate the whole regularization solution path of the mask parameter, whose support set identifies the network structure. Therefore, the solution path identifies a Transformer weight family with various sparsity levels, offering greater flexibility and customization. In this paper, we introduce such an effective pruning method, termed SPP (Solution Path Pruning). To achieve effective pruning, we segment the transformers into paired modules, including query-key pairs, value-projection pairs, and sequential linear layers, and apply low-rank compression to these pairs, maintaining the output structure while enabling structural compression within the inner states. Extensive experiments conducted on various well-known transformer backbones have demonstrated the efficacy of SPP. Our code is available at https://github.com/yizhuoDi/Solution-Path-Pruning.

## 1 INTRODUCTION

Transformers have succeeded in various tasks due to their scalability, parallel processing abilities, and capacity to learn complex data patterns. Pretrained transformers, in particular, perform well on downstream tasks by leveraging large datasets during training. These models are highly effective for transfer learning, allowing fine-tuning for different tasks and delivering strong performance in many natural language processing applications. However, their large size, often with billions of parameters, makes them difficult to deploy on low-cost hardware. Despite their widespread use and impressive performance (Radford et al., 2021; Touvron et al., 2021a), running transformers on lightweight devices like phones and laptops remains challenging due to computational constraints. Therefore, compressing transformers to run efficiently on affordable hardware is crucial.

Various techniques have been developed to compress transformers while preserving their performance. These include weight sharing (Lan et al., 2019), low-rank factorization (Yu et al., 2017), quantization (Gong et al., 2014; Polino et al., 2018; Tao et al., 2022), knowledge distillation (Hinton et al., 2015; Yuan et al., 2019; Touvron et al., 2020; Liu et al., 2020), and pruning (Yu & Xiang, 2023; Yang et al., 2023; Shi et al., 2023; Yin et al., 2023). Pruning methods typically involve structured pruning, wherein entire neurons or attention heads are removed based on importance, followed by fine-tuning to regain accuracy. Despite the array of pruning methods developed for transformers, current algorithms face a fundamental challenge: they are tailored to achieve a predefined pruning ratio by the end of the pruning phase.

Typically, when targeting a new degree of sparsity, the entire pruning process is restarted to meet the new target. Restarting such an entire pruning process for each level of sparsity introduces high costs for model compression. For over-parameterized models, the training cost can be substantial.

---

✉Corresponding authors. Prof. Yanwei Fu is also with Fudan ISTBI–ZJNU Algorithm Centre for Brain-inspired Intelligence, Zhejiang Normal University, Jinhua, China.

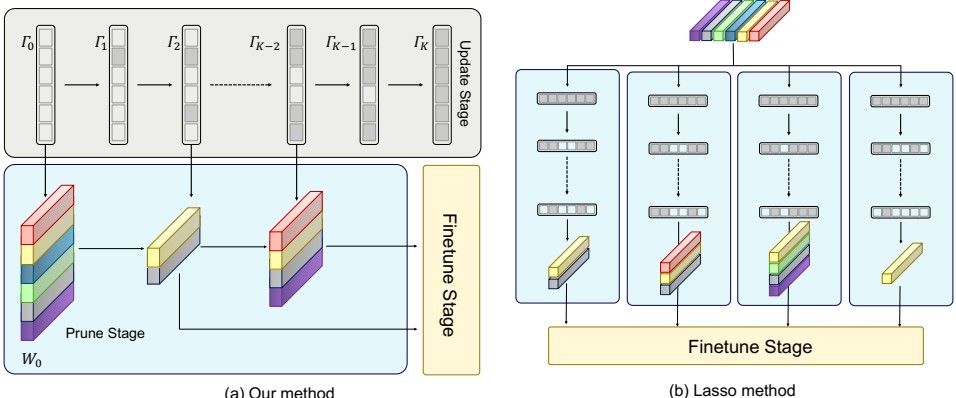

Figure 1: Comparison of SPP and lasso method. (a) SPP can obtain sparse models of all sparsity after search stage , which includes update stage and prune stage. (b) Lasso method can only obtain one sparse model in a single search stage.

To address this challenge, we propose an adaptive compression strategy through a differential inclusion for mask-based pruning of pretrained transformer, named SPP (Solution Path Pruning). The dynamics efficiently generates a whole regularization solution path of masks,where each mask's support set captures the important weights. As is shown in Figure1, along this solution trajectory, the sparsity of the projected target model incrementally increases, with important weights learned earlier in the dynamics until convergent to the fully dense network. Besides, this dynamics has a simple iterative form to implement. Therefore, after running a whole iteration, we can obtain a *transformer weight family* with different compression ratios. Note that Fu et al. (2020) used a similar pruning method for structured sparsity in neural networks trained from scratch. They also used differential inclusions with inverse scale spaces to train the network. Our focus, however, is on pre-trained transformers, which are much harder to prune while maintaining performance. Unlike common pruning methods, SPP does not require restarting the search stage, rendering it a cost-effective pruning technique requiring just one search stage to derive a transformer weight family with diverse sparsity levels from the uncompressed model.

Our exploration of transformer structures adopts a novel and efficient approach, automatically learning compression structures throughout training, thus enjoying greater flexibility and customization. Currently, no other algorithm can theoretically guarantee optimization convergence while producing sparse models with different levels of sparsity during training. For example, the Upop (Shi et al., 2023) algorithm only makes the model structure sparse at the final step, with the mask becoming zero only at that point. The OFA (Cai et al., 2019) method lacks a convergence guarantee and simply fixes the mask updates during training to achieve a sparse network. SPP addresses this by using the solution path to obtain sparse structures without affecting convergence, providing a stronger theoretical foundation. Notably, our method can be extended to prune large language models (LLMs) with one-shot post training.

In this paper, we applied SPP to the classification task dataset ImageNet (Deng et al., 2009) using the DeiT (Touvron et al., 2021a) backbone. Furthermore, we also extended the method to image and text retrieval datasets, using CLIP models (Radford et al., 2021). Our contributions are summarized as follows:

- We develop a differential inclusion-based method for adaptive compression of pretrained transformer, enabling the acquisition of sparse models with different sparsity within a single search stage, significantly reducing the cost of model pruning.
- We introduce the novel concept of the Transformer Weight Family, obtained through a simple iterative algorithm following the discretization of the differential inclusion.
- We also prove the global convergence of the method in such a non-convex optimization based on the Kurdyka-Łojasiewicz framework, demonstrating that the entire iterative sequence converges to a critical point of the empirical loss function from arbitrary initializations.
- We demonstrate the effectiveness of our framework across various backbones and datasets. Results show that we can significantly reduce the computational costs while preserving the prediction accuracy.

## 2 RELATED WORK

**Transformer pruning.** In the expanding field of Transformer-based model research, the concept of pruning Transformer has garnered considerable interest for its potential to streamline model architectures.

Over recent years, various structured pruning methods have been developed. ViT-Slim (Chavan et al., 2022) employs a single-shot training approach to search for optimal sub-structures in vision transformers. SAViT (Chen et al., 2021) collaboratively prunes all components of Vision Transformers, integrating structure-aware interactions and using a Taylor-based optimization function for joint importance. UP-ViTs (Yu & Wu, 2023) introduces a unified pruning framework for vision transformers, focusing on comprehensive pruning of all components of ViT models and their variants.

The process of pruning a Transformer model is twofold: first targeting the MLP and then the Attention mechanism. Approaches such as WDpruning (Yu et al., 2022) employ a mask-based technique. Specifically, a mask $M$ is defined to correspond to each column of the MLP's weight matrix, and pruning is conducted by considering the gradient magnitudes of this mask. However, due to the complicated structure of the Attention mechanism, such strategies may fail.

WDpruning (Yu et al., 2022) extends its paradigm by incorporating head-level pruning within the multi-head attention framework. Concurrently, methodologies like SAVIT and Upop advocate for the pruning of the input projection matrices, retaining the structural integrity of the query, key, and value matrices. Those approaches, however, lack flexibility and expandability. Because the matrices for the query, key, and value play distinct roles during the forward pass of the attention process.

Our methodology introduces an innovative approach to pruning within the attention paradigm. It enables an asymmetric dimensionality between the query, key, and value matrices after pruning, allowing for a more nuanced and efficient pruning process. This novel technique does not necessitate the uniform dimensionality across these matrices, thereby enhancing the pruning mechanism's flexibility and applicability to diverse Transformer-based architectures.

**Mirror Descent Algorithm and Bregman Inverse Scale Space.** Mirror Descent Algorithm (MDA) was first proposed by Nemirovskij & Yudin (1983) to solve constrained convex optimization, and can be seen as a generalized projected gradient descent (Beck & Teboulle, 2003) using Bregman distance $B_\Omega(u, v) := \Omega(u) - \Omega(v) - \langle \nabla\Omega(v), u - v \rangle$ induced by a convex and differentiable function $\Omega(.)$.

Convergence analysis for convex loss has been extended to stochastic versions (Ghadimi & Lan, 2013) and Nesterov acceleration (Krichene et al., 2015; Su et al., 2016). For non-convex loss in deep learning, convergence to global optima for overparameterized networks has been established (Azizan et al., 2019). For non-differentiable penalties, such as $\ell_1$ penalty for sparse recovery, Osher et al. (2016a); So et al. (2008) proposed Linearized Bregman Iteration (LBI), follows a discretized *solution path* of differential inclusions called Bregman Inverse Scale Space (ISS). These solution paths enjoy the inverse scale space property, which means important features such as the signal, will be learned earlier than non-important ones such as noise.

**Mask-based pruning for pretrained CNN.** Fu et al. (2020; 2022); Bungert et al. (2022) applied LBI to forwardly train a network from scratch, based on the lottery ticket hypothesis. By incorporating several training techniques tailored to the network architecture, the sparse network achieved comparable results to the fully dense network. **In contrast**, we propose a new differential inclusion for mask-based pruning, which can adaptively prune a pre-trained transformer. This method generates an iterative solution path that uncovers key sparse architectures early during training. Our method can perform consistently well across various backbones and datasets. Unlike ADMM (Wahlberg et al., 2012; Boyd et al., 2011), which focuses on convergence, our differential inclusion dynamics aim at generating a whole solution path with various compression ratios.

## 3 METHOD

**Problem setup.** Given the model weights $W$, inputs $X$, and the objective $\mathcal{L}$, the target of pruning is minimizing the size of model and keeping the performance of the model. i.e.

$$\min_W \mathcal{L}(X, W) \quad s.t. \quad \rho(W) < \rho, \tag{1}$$

where $\rho(W)$ is the sparsity level of model, $\rho \in (0, 1]$ is the desired degree of sparsity. Straightforward unstructured pruning simply discards unimportant weights in the architecture, but this approach often fails to meet the requirements for accelerating computation or reducing memory costs. In contrast, structural pruning reduces the complexity and computational cost of neural networks by removing entire structural units, making it more suitable for hardware acceleration and practical deployment. In this paper, we focus on the problem of structural pruning.

**Transformer weight family.** Our goal is to efficiently obtain a family of neural networks with different sparsity levels. To achieve this, we propose a dynamic approach based on differential inclusion induced by $\ell_1$ penalty. This dynamic can generate a whole regularization solution path from sparse to dense, with important weights learned earlier. Besides, it enjoys a simple iterative form to implement, which identifies a weight family with different sparsity levels. Such a weight family can be obtained from a single search stage, eliminating the need to restart the search process. This method is cost-effective compared to previous pruning methods.

## 3.1 MASK-BASED PRUNING

**Structural weight importance mask.** Without loss of generality, suppose the weight matrix is $W \in \mathbb{R}^{m \times d}$ where $d$ is the feature dimension we want to prune. We introduce the mask matrix $M = (mask_1, \ldots, mask_d) \in [0, 1]^{1 \times d}$ for every weight matrix of the transformer models with the same matrix size, where $mask_i$ indicates the importance of corresponding column. This column-wise design naturally achieves structural pruning by discarding columns of the weight matrices. Subsequently, the masked network is defined by:

$$\bar{\mathcal{L}}(W, M) = \mathcal{L}(W \odot M), \tag{2}$$

where $\odot$ denotes the Hadamard product. Our target for the mask-based structural pruning task is to learn the $M$ to identify the important columns while discarding the others.

**Pair-wise shared mask.** In transformers, adjacent layers are typically coupled in the feature dimensions. When considering them separately, manual efforts like padding are required to avoid dimension mismatch issues. To address this, many pruning approaches use a shared mask for an entire module, such as attention layers. While this ensures dimensional consistency within the same layer, it is conservative and lacks the potential for more fine-grained pruning.

To achieve both dimension matching and pruning flexibility, we propose pruning at the smallest pair-wise level. This approach maximizes flexibility without causing dimension mismatches. Specifically, transformers are primarily based on multi-head self-attention (MHSA) layers and feed-forward MLPs. We suggest dividing MHSA into query-key and value-output pairs while considering the adjacent linear layers in MLPs. Given the standard MHSA as,

$$A_i = \text{softmax}\left(X W_{Q,i}\left(X W_{K,i}\right)^T / \sqrt{d}\right), \quad V_i = A_i(X W_{V,i}), \quad O_{\text{MHSA}} = \sum V_i W_{proj,i}, \tag{3}$$

where $i$ is the $i$-th head, $W_{K,i} \in \mathbb{R}^{m \times d}$, $W_{Q,i} \in \mathbb{R}^{m \times d}$, $W_{V,i} \in \mathbb{R}^{m \times d_1}$, $W_{proj,i} \in \mathbb{R}^{d_1 \times m}$ are weight matrices of the Query, Key, Value and output projection, respectively. The $m$ and $d$ denotes the dimension of the input features and the hidden feature, respectively.

Our proposed pair-wise shared mask introduces the weight importance mask via,

$$W_{Q,i}, W_{K,i} \to W_{Q,i} \odot M_{QK}, W_{K,i} \odot M_{QK}, \tag{4a}$$

$$W_{V,i}, W_{proj,i} \to W_{V,i} \odot M_V, \to M_V^T \odot W_{proj,i} \tag{4b}$$

where $M_{QK} \in [0, 1]^{1 \times d}, M_V \in [0, 1]^{1 \times d_1}$ and $\|M_{QK}\|_0 \ll d, \|M_V\|_0 \ll d_1$. The same mask matrix are shared in the query-key pair and in value-output pair. Similarly, for the feedforward module,

$$\text{FFN}(X) = \phi\left(X W_{\text{input}}\right) W_{\text{output}}, \quad W_{\text{input}} \in \mathbb{R}^{n \times d}, W_{\text{output}} \in \mathbb{R}^{d \times n}, \phi \text{ is activate function,} \tag{5}$$

we assign a shared mask $M_{\text{MLP}} \in \mathbb{R}^{1 \times d}$ to prune $W_{\text{input}}$ and $W_{\text{output}}$:

$$W_{\text{input}}, W_{\text{output}} \to W_{\text{input}} \odot M_{\text{MLP}}, M_{\text{MLP}}^T \odot W_{\text{output}} \tag{6}$$

**Sparse optimization of masks.** To minimize information loss in the pruned model, a common objective is to ensure that the weights being pruned gradually approach zero during the search phase.

---

**Algorithm 1** Transformer Weight Family

---

Perform searching
**Input:** Pretrained weight $W_0$ and a step size $\alpha$, iteration steps in the update stage $T_s$ and prune stage $T_p$
Initialize sub-gradient $V_0 = 0$, mask $M_0 = 1$, sparse mask $\Gamma_0 = 0$.
**for** $k = 0$ **to** $T_s$ **do**
   # Calculate the loss
   $\hat{L} = L(W_0 \odot M_k) + \frac{1}{2\nu}\|M_k - \Gamma_k\|_2^2$
   # update $V_k$ and mask $M_k$ according to sub-gradient
   $M_{k+1} = M_k - \kappa\alpha_k\nabla_{M_k}\hat{L}$
   $V_{k+1} = V_k - \alpha_k\nabla_{\Gamma_k}\hat{L}$
   # update $\Gamma_k$ as the proximal operator with penalty $h(\Gamma) = \lambda\|\Gamma\|_1 + I_{[0,1]}(\Gamma)$
   $\Gamma_{k+1} = \text{Prox}_h(V_{k+1})$, where $\text{Prox}(V) = argmin_\Gamma\left\{\|V - \Gamma\|_2^2/2 + h(\Gamma)\right\}$
**end for**
Perform pruning
**for** $k = 0$ **to** $T_p$ **do**
   # Update masks with a reverse turn of $\Gamma$ being non-zero in searching
   $M_{k+1} = \Gamma_{\hat{k}}, \hat{k} = int(T_s - (k+1)\frac{T_s}{T_p})$
   # Update and save the sparse model weights
   $\bar{W}_{k+1} = W_{k+1} \odot M_{k+1}$
   #Save the checkpoint of $\bar{W}_{k+1}$ as the pruned model
**end for**
# Return Weight Family for the model
**Output:** Transformer Weight Family:$\{\bar{W}_i | i \in [0, T_p]\}$

---

To achieve this, it is essential to add a regularization term, specifically, the $L_1$ loss of the mask, to the loss function in the optimization algorithm. The updated loss function can be formulated below:

$$\bar{\mathcal{L}}(W, M) = \mathcal{L}(W \odot M) + \lambda\|M\|_1 + I_{[0,1]}(M), \tag{7}$$

where the indicator function of interval $[0, 1]$ restrict the mask to a meaningful range. And we initialize $M$ with all-one values. However, the Lasso optimization method is insufficient for adaptive compression. This limitation arises because the solution path of Lasso only exists when the optimization function is linear. As a result, Lasso can only produce a model with a certain level of sparsity, dependent on the hyperparameters. Therefore, we need to find an alternative method to effectively address the sparsification problem.

## 3.2 DIFFERENTIAL INCLUSION FOR REGULARIZATION WEIGHT FAMILY

Given pre-trained weights $W_0$, we aim to obtain a weight family with different sparsity. To effectively achieve this, we consider the following dynamics:

$$\frac{\dot{M}_t}{\kappa} = -\nabla_M\mathcal{L}_\rho(M_t, \Gamma_t), \tag{8a}$$

$$\dot{V}_t = -\nabla_\Gamma\mathcal{L}_\rho(M_t, \Gamma_t), \tag{8b}$$

$$V_t \in \partial\Omega(\Gamma_t), \tag{8c}$$

where $V$ is a sub-gradient of $\Omega(\Gamma) := \|\Gamma\|_1 + \mathbb{1}_{[0,1]}(\Gamma) + \frac{1}{2\kappa}\|\Gamma\|^2$; and $\kappa$ is a damping factor to ensure the right continuity of the solution path. Since $M$ cannot be initialized to zeros, we introduce an augmented parameter $\Gamma$ and enforce it to be sparse. To learn important weights, we also enforce $\Gamma$ to be close to $M$ through an $\ell_2$ penalty in the following appended loss:

$$\mathcal{L}_\rho(M, \Gamma) = \mathcal{L}(W_0 \odot M) + \frac{\rho}{2}\|M - \Gamma\|_F^2, \ (\rho > 0). \tag{9}$$

**Remark 1.** *Eq. 8 is a special form of a mirror descent flow with $\Omega(\Gamma)$. Unlike previous works for mirror descent primarily concerned with convergence analysis, our main objective is to obtain a regularization solution path, where each solution along this solution corresponds to a sparse mask that defines the network's sparse structure.*

The differential inclusion in Eq. 8 generates a solution path for $(M_t, \Gamma_t)$, where $M_t$ follows a gradient descent flow to learn the activation values for the pre-trained weights $W_0$, while $\Gamma_t$ is updated via

a mirror descent flow with the penalty $\Omega(\Gamma)$ to explore the sparse structure. During the updating process of $\Gamma_t$, important weights are learned earlier than non-important ones. Essentially, this is known as the inverse scale space property in inverse problems (Burger et al., 2006; Osher et al., 2016b).

Similarly, starting from $\Gamma_0 = \mathbf{0}$, $\Gamma_t$ identifies a family of network structures, where important weights will be learned in earlier epochs. Specifically, note that for each element $i$, $|V(i)| \leq 1$, if $\Gamma(i) = 0$, and is equal to 1 when it becomes non-zeros. Therefore, driven by the gradient descent in Eq. 8b that gives a dense solution $\lim_{t \to \infty}(M_t, \Gamma_t) = \arg\min_{M,\Gamma} \mathcal{L}_\rho(M, \Gamma)$, $V_t$ that begins from $V_0 = \mathbf{0}$, will have more of its elements reaching the boundary of 1, making corresponding elements in $\Gamma_t$ becoming non-zeros. Compared to directly solving Eq. 7, our dynamics is more efficient in generating a network family with various sparsity levels and, hence is more flexible for deployment.

**Remark 2.** *The differential inclusion has been previously explored to forwardly train a sparse network from scratch (Fu et al., 2020; 2022; Bungert et al., 2022), incorporating several deep learning techniques tailored to the multilayer perceptron and the convolutional neural network. However, these techniques may not be applicable to the Transformer, and thus may lead to suboptimal performance. In contrast, our dynamics involves backward pruning of a pre-trained network. Furthermore, instead of weight-based pruning, we employ mask-based pruning, which is significantly easier to train.*

To implement, we consider the following iteration, beginning from $V_0 = \Gamma_0 = \mathbf{0}$, and $M_0 = \mathbf{1}$:

$$M_{k+1} = M_k - \kappa\alpha_k \cdot \nabla_M \mathcal{L}_\rho(M_k, \Gamma_k), \tag{10a}$$

$$V_{k+1} = V_k - \alpha_k \cdot \nabla_\Gamma \mathcal{L}_\rho(M_k, \Gamma_k), \tag{10b}$$

$$\Gamma_{k+1} = \kappa \cdot \text{Prox}_\Omega(V_{k+1}), \tag{10c}$$

where $\alpha$ is the step size and should be small enough to well approximate the original dynamics in Eq. 8. In Eq. 10b, the proximal operator $\text{Prox}_\Omega(\cdot)$ is defined as:

$$\text{Prox}_\Omega(V) = \arg\min_\Gamma \left\{ \frac{1}{2}\|\Gamma - V\|^2 + \Omega_(\Gamma) \right\}, \tag{11}$$

which gives $\Gamma(i) = 1$ if $V(i) > 2$, $= V(i) - 1$ if $1 < V \leq 2$, $= 0$ otherwise. Running Eq. 10 will obtain a solution path of $\Gamma_k$ with increasing levels, that define a family of weights dubbed as *Transformer weight family*:

$$W_k = W_0 \odot \Gamma_k. \tag{12}$$

# 4 CONVERGENCE

We present a theorem that guarantees the *global convergence* of our method, *i.e.* from any initialization, the sequence converges to a critical point of $\bar{\mathcal{L}}$. Our treatment extends the Block Coordinate Descent (BCD) studied in Zeng et al. (2019), with a crucial difference being the mirror descent involved in our method. Instead of the splitting loss in BCD, a new Lyapunov function is developed here to meet the Kurdyka-Łojasiewicz property (Łojasiewicz, 1963). Let $P := (M, \Gamma)$. Following Huang & Yao (2018), our method algorithm can be rewritten as the following standard linearized Bregman iteration,

$$P_{k+1} = \arg\min_P \left\{ \langle P - P_k, \alpha\nabla\bar{\mathcal{L}}(P_k) \rangle + B_\Psi^{p_k}(P, P_k) \right\} \tag{13}$$

where

$$\Psi(P) = \Omega_\lambda(\Gamma) + \frac{1}{2\kappa}\|P\|_2^2 = \Omega_\lambda(\Gamma) + \frac{1}{2\kappa}\|W\|_2^2 + \frac{1}{2\kappa}\|\Gamma\|_2^2,$$

$p_k \in \partial\Psi(P_k)$, and $B_\Psi^q$ is the Bregman divergence associated with convex function $\Psi$, defined by

$$B_\Psi^q(P, Q) := \Psi(P) - \Psi(Q) - \langle q, P - Q \rangle. \tag{14}$$

for some $q \in \partial\Psi(Q)$. Without loss of generality, consider $\lambda = 1$ in the sequel. One can establish the global convergence of our method under the following conditions.

**Condition 1.** *Suppose that: (a) $L(W) = \frac{1}{n}\sum_{i=1}^n \ell(y_i, \Phi_W(x_i))$ is continuous differentiable and $\nabla L$ is Lipschitz continuous with a positive constant Lip; (b) $L(W)$ has bounded level sets; (c) $L(W)$ is lower bounded (without loss of generality, we assume that the lower bound is 0); (d) $\Omega$ is a proper*

*lower semi-continuous convex function and has locally bounded subgradients, that is, for every compact set $\mathcal{S} \subset \mathbb{R}^n$, there exists a constant $C > 0$ such that for all $\Gamma \in \mathcal{S}$ and all $g \in \partial\Omega(\Gamma)$, there holds $\|g\| \leq C$; and (e) the Lyapunov function*

$$F(P, \tilde{g}) := \alpha\bar{\mathcal{L}}(M, \Gamma) + B_{\Omega}^{\tilde{g}}(\Gamma, \tilde{\Gamma}), \tag{15}$$

*is a Kurdyka-Łojasiewicz function on any bounded set, where $B_{\Omega}^{\tilde{g}}(\Gamma, \tilde{\Gamma}) := \Omega(\Gamma) - \Omega(\tilde{\Gamma}) - \langle \tilde{g}, \Gamma - \tilde{\Gamma} \rangle$, $\tilde{\Gamma} \in \partial\Omega^*(\tilde{g})$, and $\Omega^*$ is the conjugate of $\Omega$ defined as*

$$\Omega^*(g) := \sup_{U \in \mathbb{R}^n} \{\langle U, g \rangle - \Omega(U)\}.$$

Now we are ready to present the main theorem.

**Theorem 1.** *[Global Convergence of SPP] Suppose that Assumption 1 holds. Let $(W_k, \Gamma_k)$ be the sequence generated by Our method (Eq. (10)) with a finite initialization. If*

$$0 < \alpha_k = \alpha < \frac{2}{\kappa(Lip * C + \nu^{-1})}, \tag{16}$$

*where $C = \max|W_0|$ is a max value of the pretrained model then $(M_k, \Gamma_k)$ converges to a critical point of $\bar{\mathcal{L}}$ defined in Eq. (10), and $\{M^k\}$ converges to a critical point of $\mathcal{L}(W)$.*

## 5 EXPERIMENTS

In this section, we evaluated our method on three transformer models. These models were trained on two classical datasets, ImageNet-1k, COCO and a smaller dataset, CIFAR-10. We also conducted ablation studies to highlight the importance of introducing the mask parameters and gamma buffer. Additionally, we extended our method to large language models (LLMs) such as Llama2-7b and OPT-6.7b. More experimental details are provided in Table 6 and Table 8.

**Experimental setting.** Our method begins with a pretrained model, keeping all parameters unchanged except for the mask parameters. The search stage is performed only once for all degrees of sparsity. From the solution path, we derive a sparse transformer weight family by applying early stopping and saving the model checkpoint when the desired sparsity is achieved. As described in Algorithm 1, each model produces a transformer weight family with a sparse architecture.

### 5.1 MAIN RESULTS

We reported our results pruning DeiT, Swin and CLIP model with various degrees of sparsity.

**Transformer weight family results of DeiT.** Table 1 shows our transformer weight family results of DeiT on ImageNet-1k. The results show that our proposed method performed well on DeiT. For DeiT-Small, our method maintained accuracy at 80.2% while reducing 29.5% of the parameters. Compared to the other methods, WDpruning only achieved 78.6% accuracy with a 29.6% compression ratio. Only UPop (Shi et al., 2023) achieved comparable results to ours. This indicates that our method of solution path is highly effective, preserving more parameter information even at higher compression ratio.

Our method performed well on DeiT-Base. At almost the same compression ratio, our accuracy surpassed SCOP (Tang et al., 2020) and PoWER (Goyal et al., 2020), and we achieved a better compression ratio than IA-RED (Pan et al., 2021), the smallest model among the compared algorithms.

**Transformer weight family results of CLIP.** Apart from the classification task, we applied our method to image and text retrieval tasks. Using the CLIP backbone, we compress the transformer module to different sparsity levels, as shown in Table 3. Our method is robust, maintaining high recall under different compression ratios. Notably, our method excels in Image-to-Text retrieval, with only a 1% performance drop while using around 60% of the FLOPs of the full models.

SPP also demonstrates its capability for low-cost pruning. In the CLIP case, after a 6-epoch search stage, we obtained 5 sparse model architectures with different sparsity levels. Each sparse model only needs to be retrained for 5 epochs to achieve good performance.

Table 1: The results of DeiT(Touvron et al., 2021b) models.We compared the results with the other advanced methods.The reduce is the reduce of parameters.

| Model | Method | Top-1(%) | Top-5(%) | FLOPS(B) | Params(M) |
|---|---|---|---|---|---|
| DeiT-Small | Uncompressed | 79.8 | 95.0 | $4.6_{100\%}$ | $22.1_{100\%}$ |
| | $S^2$ ViTE-Small | 79.2 | - | - | $14.6_{66.1\%}$ |
| | GOHSP | 80.0 | - | $3.0_{65.2\%}$ | $14.4_{65.2\%}$ |
| | PS-ViT-S | 79.4 | - | $2.7_{58.7\%}$ | $22.0_{99.5\%}$ |
| | ViTAS - E | 77.4 | 93.8 | $2.7_{58.7\%}$ | $12.6_{57.0\%}$ |
| | Upop | 79.6 | 94.8 | $2.8_{60.9\%}$ | $13.5_{61.1\%}$ |
| | Upop | 80.2 | 95.1 | $3.2_{69.6\%}$ | $15.7_{71.0\%}$ |
| | ViT-Slim | 80.0 | 95.1 | $3.3_{71.7\%}$ | $15.7_{71.0\%}$ |
| | WDPruning | 78.6 | 94.4 | $3.1_{67.4\%}$ | $15.0_{67.9\%}$ |
| | WDPruning | 78.4 | 94.1 | $2.6_{56.5\%}$ | $13.3_{60.2\%}$ |
| | X-Pruner | 78.9 | 94.2 | $2.4_{52.2\%}$ | - |
| | OPTIN | 79.2 | - | $3.2_{68.4\%}$ | - |
| | **SPP** | **80.2** | **95.1** | $3.4_{73.9\%}$ | $15.6_{70.6\%}$ |
| | **SPP** | **78.9** | **94.6** | $2.6_{56.5\%}$ | $12.6_{57.0\%}$ |
| | **SPP** | **77.7** | **94.0** | $1.9_{41.3\%}$ | $10.4_{47.1\%}$ |
| DeiT-Tiny | Uncompressed | 72.2 | 91.1 | 1.3 | 5.7 |
| | GOHSP | 70.2 | - | $0.9_{69.2\%}$ | $4.0_{70.2\%}$ |
| | $S^2$ViTE | 70.1 | - | $1.0_{76.9\%}$ | $4.2_{73.7\%}$ |
| | WDPruning | 70.3 | 89.8 | $0.7_{53.8\%}$ | - |
| | PoWER | 69.4 | 89.2 | $0.8_{61.5\%}$ | - |
| | UPDP | 70.3 | - | $0.9_{70.3\%}$ | $3.8_{66.7\%}$ |
| | MCF | 71.5 | - | $0.7_{53.8\%}$ | $3.9_{68.4\%}$ |
| | OPTIN | 71.3 | - | $0.9_{70.3\%}$ | - |
| | **SPP** | **72.3** | **91.1** | $0.8_{61.5\%}$ | $4.0_{70.2\%}$ |
| DeiT-Base | Uncompressed | 81.8 | 95.6 | $17.5_{100\%}$ | $86.6_{100\%}$ |
| | ViT-B/16 | 77.9 | 95.3 | $17.5_{100\%}$ | $86.6_{100\%}$ |
| | SCOP | 79.7 | 94.5 | $10.2_{58.3\%}$ | $58.3_{67.3\%}$ |
| | IA-RED | 80.3 | - | $11.8_{67.4\%}$ | $67.0_{77.4\%}$ |
| | PoWER | 80.1 | 94.6 | $10.4_{59.4\%}$ | - |
| | X-Pruner | 81.02 | 95.38 | $8.5_{48.6\%}$ | - |
| | **SPP** | **81.9** | **95.7** | $9.8_{56.0\%}$ | $48.1_{55.5\%}$ |
| | **SPP** | **81.2** | **95.4** | $6.9_{39.4\%}$ | $34.2_{39.5\%}$ |
| | **SPP** | **78.1** | **93.8** | $4.4_{25.1\%}$ | $22.0_{25.4\%}$ |

Table 2: Results of Swin-Tiny and DeiT-Tiny on Imagenet-1k, together with result of evaluating DeiT-Small on Cifar10.

| Model | Method | T-1(%) | T-5(%) | FLOPS(B) | Pa. (M) |
|---|---|---|---|---|---|
| Swin-Tiny | Uncompressed | 81.2 | 95.5 | 4.5 | 28.0 |
| | ViT-Slim | 80.7 | 95.4 | $3.4_{75.6\%}$ | $19.4_{69.3\%}$ |
| | **SPP** | **80.6** | **95.2** | $3.4_{75.6\%}$ | $18.5_{66.1\%}$ |
| DeiT-Tiny | Uncompressed | 72.2 | 91.1 | 1.3 | 5.7 |
| | GOHSP(Yin et al., 2023) | 70.2 | - | $0.9_{69.2\%}$ | $4.0_{70.2\%}$ |
| | $S^2$ViTE(Chen et al., 2021) | 70.1 | - | $1.0_{76.9\%}$ | $4.2_{73.7\%}$ |
| | WDPruning(Yu et al., 2022) | 70.3 | 89.8 | $0.7_{53.8\%}$ | - |
| | PoWER(Goyal et al., 2020) | 69.4 | 89.2 | $0.8_{61.5\%}$ | - |
| | **SPP** | **72.3** | **91.1** | $0.8_{61.5\%}$ | $4.0_{70.2\%}$ |
| DeiT-Small | Uncompressed | 98.5 | - | 4.6 | 22.1 |
| | ViT-Slim(Chavan et al., 2022) | 98.7 | - | $3.3_{71.7\%}$ | $15.6_{70.6\%}$ |
| | WDPruning(Yu et al., 2022) | 98.1 | - | $2.8_{60.9\%}$ | $14.9_{67.4\%}$ |
| | **SPP** | **98.8** | - | $3.3_{71.7\%}$ | $15.4_{69.7\%}$ |

Table 3: Ablation: The results of CLIP-large and CLIP-base.

| Model | Method | Image->Text | | | Text->Image | | | Params(M) | FLOPS(B) |
|---|---|---|---|---|---|---|---|---|---|
| | | $R@1$ | $R@5$ | $R@10$ | $R@1$ | $R@5$ | $R@10$ | | |
| CLIP-Large | Uncompressed | 71.5 | 90.8 | 95.4 | 56.8 | 80.7 | 87.6 | $856.0_{100\%}$ | $395.7_{100\%}$ |
| | | 73.7 | 92.5 | 96.2 | 55.6 | 79.1 | 85.7 | $807_{94.3\%}$ | $376.8_{95.2\%}$ |
| | | 71.9 | 91.6 | 95.6 | 55.5 | 79.3 | 86.3 | $757_{88.4\%}$ | $353.3_{89.3\%}$ |
| | **SPP** | 70.4 | 90.7 | 95.3 | 55.5 | 80.6 | 87.8 | $699_{81.7\%}$ | $324.8_{82.1\%}$ |
| | | 70.8 | 90.9 | 95.5 | 54.6 | 80.1 | 87.4 | $650_{75.9\%}$ | $299.3_{75.6\%}$ |
| | | 70.3 | 90.5 | 95.3 | 52.5 | 78.8 | 86.4 | $532_{62.1\%}$ | $245.1_{61.9\%}$ |
| CLIP-Base | Uncompressed | 52.5 | 76.4 | 84.2 | 33.0 | 57.9 | 68.7 | $299_{100\%}$ | $41.2_{100\%}$ |
| | | 69.0 | 89.6 | 94.8 | 84.5 | 78.5 | 86.7 | $278_{93.0\%}$ | $38.5_{93.4\%}$ |
| | | 65.9 | 88.6 | 94.1 | 50.0 | 77.3 | 86.1 | $257_{86.0\%}$ | $35.5_{86.2\%}$ |
| | **SPP** | 61.9 | 86.4 | 93.1 | 47.0 | 75.6 | 84.9 | $234_{78.3\%}$ | $31.9_{77.4\%}$ |
| | | 49.8 | 78.5 | 87.3 | 35.4 | 65.5 | 77.2 | $181_{60.5\%}$ | $22.9_{55.6\%}$ |
| | | 31.4 | 61.3 | 73.7 | 21.2 | 49.1 | 62.5 | $130_{43.5\%}$ | $13.6_{33.0\%}$ |

Table 4: Comparisons with training from scratch method on pruning DeiT models.

| Model | Method | Top-1(%) | Top-5(%) | FLOPS(B) | Params(M) |
|---|---|---|---|---|---|
| DeiT-Small | Uncompressed | 79.8 | 95.0 | $4.6_{100\%}$ | $22.1_{100\%}$ |
| | DessiLBI | 78.9 | 94.2 | $3.2_{69.6\%}$ | $15.2_{68.8\%}$ |
| | **SPP** | **80.2** | **95.1** | $3.4_{73.9\%}$ | $15.6_{70.6\%}$ |
| DeiT-Tiny | Uncompressed | 72.2 | 91.1 | $1.3_{100\%}$ | $5.7_{100\%}$ |
| | DessiLBI | 71.8 | 90.8 | $1.0_{76.9\%}$ | $4.0_{70.2\%}$ |
| | **SPP** | **72.3** | **91.1** | $0.8_{61.5\%}$ | $4.0_{70.2\%}$ |
| Swin-Tiny | Uncompressed | 81.2 | 95.5 | $4.5_{100\%}$ | $28_{100\%}$ |
| | DessiLBI | 80.4 | 95.2 | $3.4_{75.6\%}$ | $18.3_{65.4\%}$ |
| | **SPP** | **80.6** | **95.2** | $3.4_{75.6\%}$ | $18.5_{66.1\%}$ |

**Results of tiny models and datasets.** We also extended our method to the Tiny models like Swin-Tiny and DeiT-Tiny model. Table 2 presents a comparison between our approach and the other method, ViT-Slim (Chavan et al., 2022). Notably, our method achieves only a 0.8% reduction in accuracy while compressing 5% more than ViT-Slim (Chavan et al., 2022). The success of our method with the Swin-T model, which demonstrates its adaptability across various transformer models.

For DeiT-Tiny, at almost the same compression ratio, our accuracy surpassed SSP (Chen et al., 2021) and $S^2$ViTE (Chen et al., 2021) by around 2% points, and we achieved a better compression ratio than GOHSP (Yin et al., 2023), the smallest model among the compared algorithms.

On *CIFAR-10* dataset, as shown in Table 2, our method maintained a reasonably good accuracy even at higher compression ratio, achieving better accuracy with a smaller model than ViT-Slim, demonstrating its effectiveness across various datasets.

## 5.2 ABLATION STUDIES

**Ablation of training from scratch method.** To highlight the improvements of our method over training from scratch, we conducted experiments using the DessiLBI method, as shown in Table 4. Both two methods finetuned from the same pretrained model. While the DessiLBI method can be applied to transformers, there was still a noticeable performance gap compared to our approach. This clearly demonstrated that our method significantly enhances the differential inclusion pruning technique.

## 5.3 FURTHER STUDIES

**The consistency among family.** We plotted the solution path of $\Gamma$ as shown in Figure 2. The $\Gamma$ parameters do not return to zero during training, which shows that the weights within the same family

Table 5: Results of pruning LLMs. We pruned the Llama2-7B and OPT-6.7B model with **50% sparsity**, then evaluated the pruned model on 6 datasets.

| Model | Method | Calib data | ARC-c(%) | ARC-e(%) | BoolQ(%) | RTE(%) | SST(%) |
|-------|--------|-----------|----------|----------|----------|--------|--------|
| Llama2-7B | Uncompressed | - | 43.52 | 76.26 | 77.71 | 62.82 | 51.95 |
| | RIA | C4 | 38.40 | 71.59 | 75.60 | 54.51 | 49.77 |
| | RIA | Wikitext2 | 37.97 | 71.68 | 75.17 | 55.96 | 50.57 |
| | Wanda | C4 | 37.03 | 69.70 | 74.01 | 55.23 | **53.10** |
| | Wanda | Wikitext2 | 37.29 | 69.65 | 74.28 | **57.04** | 51.72 |
| | **SPP** | C4 | **38.57** | **71.80** | **75.96** | 55.96 | 49.66 |
| | **SPP** | Wikitext2 | 37.88 | 71.59 | 74.28 | 54.51 | 50.11 |
| OPT-6.7B | Uncompressed | - | 30.46 | 65.57 | 66.06 | 55.23 | 76.61 |
| | RIA | C4 | **29.27** | 63.68 | **66.82** | 53.07 | 61.81 |
| | RIA | Wikitext2 | 29.18 | **64.10** | 63.55 | 52.71 | 76.49 |
| | Wanda | C4 | 27.39 | 57.45 | 63.88 | 50.90 | **78.21** |
| | Wanda | Wikitext2 | 26.11 | 56.40 | 62.20 | **53.43** | 62.96 |
| | **SPP** | C4 | 28.75 | 63.76 | 63.15 | 52.71 | 74.66 |
| | **SPP** | Wikitext2 | 28.67 | 63.80 | 63.18 | 52.71 | 75.34 |

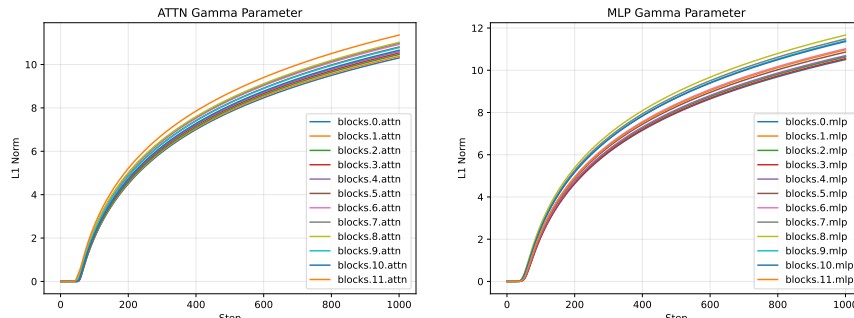

Figure 2: Visualization of solution path of DeiT-small. We show the changes of the L1-norm of projected weight value $\Gamma$ during the search stage. The x-axis is the iteration number during training, the y-axis is the L1-norm of the $\Gamma$ parameters per layer.

remain consistent. Such consistency among weights allow us to effectively analyze the performance of sparse models.

**Extentions to LLMs.** We extended our method to large language models (LLMs) with post-training. We applied our solution path method during the LLM pruning search stage, combining it with the RIA (Zhang et al., 2024) pruning metric. The detailed algorithm is shown in Algorithm 2 .

We applied our method on Llama2-7B and OPT-6.7b. The calibration datasets C4 and Wikitext2 were used to generate activations during the forward pass, which, along with weight magnitude, served as the pruning metric. The results were reported on 5 datasets. As shown in Table 5, the accuracy after pruning with 50% sparsity is comparable to the advanced method RIA and Wanda (Sun et al., 2023), indicating the potential our method in pruning LLMs.

## 6 CONCLUSION

We proposed a dynamic approach based on differential inclusion, which can adaptively prune any pre-trained transformers with various compression ratios. Along this path, a series of models, named the Transformer Weight Family, was derived from the masks in the solution path. With just a single run of iteration, we can achieve all sparsity levels of the original pre-trained model. We have demonstrated the stability and consistency of the Transformer Weight Family, showing that the solution path method is robust. We also demonstrated the potential of our method for pruning large language models (LLMs). In future work, we will explore this direction with a more tailored design suited to the architecture and post-training processes of LLMs.

## ACKNOWLEDGEMENTS

This work was supported by the Science and Technology Commission of Shanghai Municipality(No. 24511103100).

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

APPENDIX

## A  EXPERIMENTS DETAILS AND VISUALIZATION

**Experiments details**

As shown in Algorithm 2, instead of directly using our algorithm, we combined it with the RIA Zhang et al. (2024) pruning method. This is because post-training tasks are quite different from fine-tuning tasks. To maintain the generalization performance of LLMs, we need to take the size of the weight parameters and activations into account.

We used 4 A100 GPU with memory size of 80GB for those experiments. The search stage contain an update stage and prune stage, both just need to run once for one model. All the finetune stage used AdamW as the optimizer and consine scheduler as the learning rate scheduler. The hyperparameters are listed in Table. 6 and Table. 8.

| Model | Datasets | Updating Epochs | Pruning Epochs | Finetuning Epochs | Batch Size | LR |
|---|---|---|---|---|---|---|
| DeiT-Base | Imagenet-1k | 5 | 20 | 300 | 1024 | 8e-4 |
| DeiT-Small | Imagenet-1k | 10 | 30 | 300 | 512 | 8e-4 |
| DeiT-Tiny | Imagenet-1k | 10 | 30 | 300 | 256 | 8e-4 |
| Swin-Tiny | Imagenet-1k | 5 | 20 | 300 | 256 | 8e-4 |
| CLip-Large | COCO | 1 | 5 | 5 | 32 | 1e-5 |
| CLip-Base | COCO | 3 | 5 | 5 | 32 | 1e-5 |

Table 6: The hyperparameters of experiments mentioned above.

---

**Algorithm 2** Extention to LLMs

---

Perform searching

**Input:** Pretrained weight $W_0$ and a step size $\alpha$, iteration steps in the update stage $T_s$ and prune stage $T_p$

Initialize sub-gradient $V_0 = 0$, mask $M_0 = 1$, sparse mask $\Gamma_0 = 0$.

Set $\lambda = \lambda_0 \left( \frac{|\mathbf{W}_{ij}|}{\sum |\mathbf{W}_{*j}|} + \frac{|\mathbf{W}_{ij}|}{\sum |\mathbf{W}_{i*}|} \right) \times (\|\mathbf{X}_i\|_2)$,which is the pruning metric of RIA Zhang et al. (2024)

**for** $k = 0$ **to** $T_s$ **do**

  # Calculate the loss

  $\hat{L} = L(W_0 \odot M_k) + \frac{1}{2\nu}\|M_k - \Gamma_k\|_2^2$

  # update $V_k$ and mask $M_k$ according to sub-gradient

  $M_{k+1} = M_k - \kappa\alpha_k \nabla_{M_k}\hat{L}$

  $V_{k+1} = V_k - \alpha_k \nabla_{\Gamma_k}\hat{L}$

  # update $\Gamma_k$ as the proximal operator

  $h(\Gamma) = \lambda \|\Gamma\|_1 + I_{[0,1]}(\Gamma)$

  $\Gamma_{k+1} = \text{Prox}_h (V_{k+1})$, where $\text{Prox}(V) = argmin_\Gamma \left\{ \|V - \Gamma\|_2^2 /2 + h(\Gamma) \right\}$

**end for**

Perform pruning

**for** $k = 0$ **to** $T_p$ **do**

  # Update masks with a reverse turn of $\Gamma$ being non-zero in searching

  $M_{k+1} = \Gamma_{\hat{k}}, \hat{k} = int(T_s - (k+1)\frac{T_s}{T_p})$

  # Update and save the sparse model weights

  $\bar{W}_{k+1} = W_{k+1} \odot M_{k+1}$

  #Save the checkpoint of $\bar{W}_{k+1}$ as the pruned model

**end for**

# Return Weight Family for the model

**Output:** LLM Weight Family:$\{\bar{W}_i | i \in [0, T_p]\}$

---

| Model | Method | Latency Time | Params | FLOPS(B) |
|---|---|---|---|---|
| | Uncompressed | 9.273s | $299_{100\%}$ | $41.2_{100\%}$ |
| CLip-Base | SPP | 8.675s | $278_{93.0\%}$ | $38.5_{93.4\%}$ |
| | | 7.859s | $257_{86.0\%}$ | $35.5_{86.2\%}$ |
| | | 7.341s | $234_{78.3\%}$ | $31.9_{77.4\%}$ |
| | | 6.490s | $181_{60.5\%}$ | $22.9_{55.6\%}$ |
| | | 5.461s | $130_{43.5\%}$ | $13.6_{33.0\%}$ |

Table 7: The latency results of pruned model.

**Visualization of compressed model**

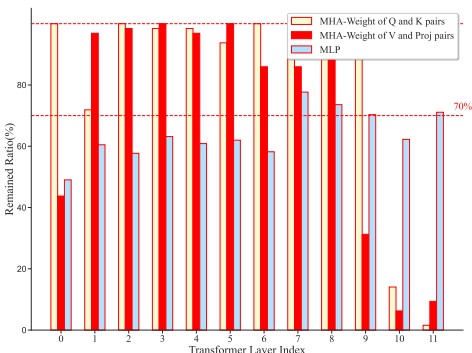

Figure 3: Visualization of the proportion of parameters on DeiT-Small. The three kind of color indicate three pairs of weight.

In Fig. 3 and Fig 4, we visualized the proportion of parameters retained in each layer of the DeiT-Small model at a set compression ratio of 0.3. Notably, since the query, key, and value, as well as the output projection, form two pairs with equivalent parameter quantities, we treated them as a unit. It's observed that in the shallower layer 1 and deeper layer 10, the proportion of saved parameters in the QK pair is significantly higher than that in the V and Project pair. This indicates that our low-rank pruning method is effective. It skillfully segregates parameters, allowing those with closer relationships to be pruned together.

## A.1 GROUP LASSO

Our algorithm 1 enhances structural sparsity within transformer layers, aligning under a group lasso penalty framework, $\Omega_1(\Gamma) = \sum_g \|\Gamma^g\|_2$, where

$$\|\Gamma^g\|_2 = \sqrt{\sum_{i=1}^{|\Gamma^g|}(\Gamma_i^g)^2} \tag{17}$$

| Model | $\kappa$ | $\lambda$ | Epochs | LR |
|---|---|---|---|---|
| DeiT-Base | 1 | 15 | 5 | 8e-4 |
| DeiT-Small | 1 | 15 | 10 | 8e-4 |
| DeiT-Tiny | 1 | 15 | 10 | 1.6e-3 |
| Swin-Tiny | 1 | 15 | 5 | 8e-4 |
| CLip-Large | 100 | 3 | 1 | 1e-5 |
| CLip-Base | 100 | 3 | 3 | 1e-5 |

Table 8: The hyperparameters of expriments mentioned above in updating stage.

| Model | Method | Latency Time | Params | FLOPS(B) |
|---|---|---|---|---|
| | Uncompressed | 9.273s | $299_{100\%}$ | $41.2_{100\%}$ |
| CLIP-base | SPP | 8.675s | $278_{93.0\%}$ | $38.5_{93.4\%}$ |
| | | 7.859s | $257_{86.0\%}$ | $35.5_{86.2\%}$ |
| | | 7.341s | $234_{78.3\%}$ | $31.9_{77.4\%}$ |
| | | 6.490s | $181_{60.5\%}$ | $22.9_{55.6\%}$ |
| | | 5.461s | $130_{43.5\%}$ | $13.6_{33.0\%}$ |

Table 9: The latency time results of compressed CLIP-baseTouvron et al. (2021b) models.

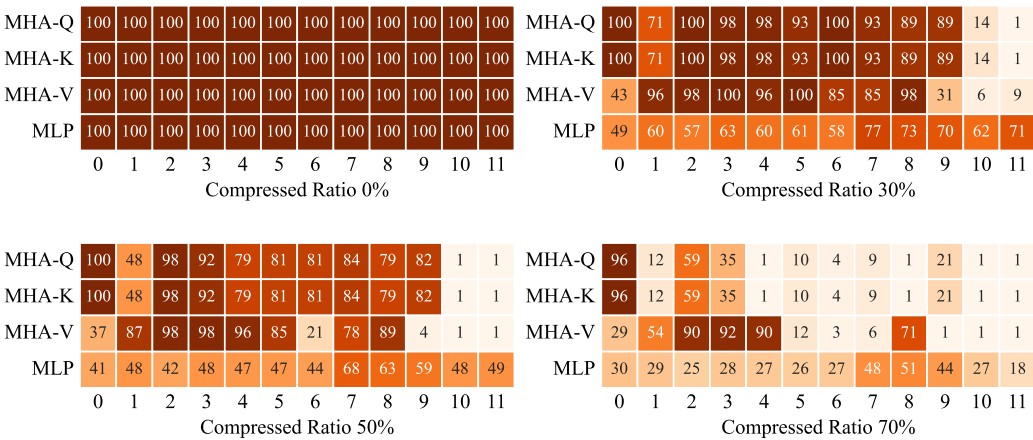

Figure 4: Visualization of components. The depth of color shows the sparsity of the corresponding layer. The number shows the dim of the linear matrix.

and $|\Gamma^g|$ represents the number of weights in each group $\Gamma^g$. Therefore, we have a clear formula for solving this equation:

$$\Gamma^g = \kappa \cdot \max(0, 1 - \frac{1}{\|\Gamma^g\|_2})V^g. \tag{18}$$

## A.2 MORE RELATED WORKS

**Mirror Descent Algorithm (MDA)** was first proposed by Nemirovskij & Yudin (1983) to solve constrained convex optimization, and can be seen as a generalized projected gradient descent Beck & Teboulle (2003) using **Bregman distance** $B_\Omega(u, v) := \Omega(u) - \Omega(v) - \langle \nabla \Omega(v), u - v \rangle$ induced by a convex and differentiable function $\Omega(.)$,

$$Z_{k+1} = Z_k - \alpha \nabla L(W_k) \tag{19a}$$
$$W_{k+1} = \nabla \Omega^*(Z_{k+1}) \tag{19b}$$

where the conjugate function of $\Omega(.)$ is $\Omega^*(Z) := \sup_{W,Z}\{\langle W, Z \rangle - \Omega(W)\}$. Equation (1) optimizes $W_{k+1}$ in two steps: Eq.19a implements the gradient descent on $Z$ in the dual space $Z_k = \nabla \Omega(W_k)$; and Eq.19b projects it back to the primal space. As step size $\alpha \to 0$, MDA converges to the following ordinary differential equation (ODE) dynamicsSu et al. (2016):

$$\dot{Z}_t = \alpha \nabla \mathcal{L}(W_t) \tag{2a}$$
$$W_t = \nabla \Omega^*(Z_t) \tag{2b}$$

**Compared with DessiLBI** Both our method and DessiLBI use the mirror descent algorithm to get the solution path, with the sparse model obtained through early stopping. However, our method is specifically designed for *pre-trained models*, while DessiLBI, which trains from scratch, doesn't perform well on them. During the search stage, we keep the pre-trained model's weights fixed and only update the mask parameters, making our approach more suitable for tasks like pruning LLMs.

Additionally, our method uses a pair-wise mask architecture, which works for fully connected layers in transformers, but is not applicable to CNN architectures.

# B PROOF OF THEOREM 1

First of all, we reformulate Eq.(13) into an equivalent form. Without loss of generality, consider $\Omega = \Omega_1$ in the sequel. Denote $R(P) := \Omega(\Gamma)$, then Eq.(13) can be rewritten as:

$$P_{k+1} = \text{Prox}_{\kappa R}(P_k + \kappa(p_k - \alpha\nabla\bar{\mathcal{L}}(P_k))), \tag{20a}$$

$$p_{k+1} = p_k - \kappa^{-1}(P_{k+1} - P_k + \kappa\alpha\nabla\bar{\mathcal{L}}(P_k)), \tag{20b}$$

where $p_k = [0, g_k]^T \in \partial R(P_k)$ and $g_k \in \partial\Omega(\Gamma_k)$. Thus Algorithm is equivalent to the following iterations,

$$M_{k+1} = M_k - \kappa\alpha\nabla_W\bar{\mathcal{L}}(M_k, \Gamma_k), \tag{21a}$$

$$\Gamma_{k+1} = \text{Prox}_{\kappa\Omega}(\Gamma_k + \kappa(g_k - \alpha\nabla_\Gamma\bar{\mathcal{L}}(M_k, \Gamma_k))), \tag{21b}$$

$$g_{k+1} = g_k - \kappa^{-1}(\Gamma_{k+1} - \Gamma_k + \kappa\alpha \cdot \nabla_\Gamma\bar{\mathcal{L}}(M_k, \Gamma_k)). \tag{21c}$$

Exploiting the equivalent reformulation (21a-21c), one can establish the global convergence of $(M_k, \Gamma_k, g_k)$ based on the Kurdyka-Łojasiewicz framework. In this section, the following extended version of Theorem 1 is actually proved.

**Theorem 2.** *[Global Convergence of Our method] Suppose that Assumption 1 holds. Let $(M_k, \Gamma_k, g_k)$ be the sequence generated by Our method (Eq. (21a-21c)) with a finite initialization. If*

$$0 < \alpha_k = \alpha < \frac{2}{\kappa(Lip * C + \nu^{-1})},$$

*then $(M_k, \Gamma_k, g_k)$ converges to a critical point of $F$. Moreover, $\{(M_k, \Gamma_k)\}$ converges to a stationary point of $\bar{\mathcal{L}}$ defined in Eq. 9, and $\{M_k\}$ converges to a stationary point of $\mathcal{L}(M)$.*

**Remark 3.** *Assumption 1 (a)-(c) are regular in the analysis of nonconvex algorithm (see, Attouch et al. (2013) for instance), while Assumption 1 (d) is also mild including all Lipschitz continuous convex function over a compact set. Some typical examples satisfying Assumption 1(d) are the $\ell_1$ norm, group $\ell_1$ norm, and every continuously differentiable penalties. By Eq. (15) and the definition of conjugate, the Lyapunov function $F$ can be rewritten as follows,*

$$F(W, \Gamma, g) = \alpha\bar{\mathcal{L}}(W, \Gamma) + \Omega(\Gamma) + \Omega^*(g) - \langle\Gamma, g\rangle. \tag{22}$$

Applying to the neural networks, typical examples are summarized in the following corollary.

**Corollary 1.** *Let $\{M_k, \Gamma_k, g_k\}$ be a sequence generated by Our method (21a-21c) for neural network training where (a) $\ell$ is any smooth definable loss function, such as the square loss $(t^2)$, exponential loss $(e^t)$, logistic loss $\log(1+e^{-t})$, and cross-entropy loss; (b) $\sigma_i$ is any smooth definable activation, such as linear activation $(t)$, sigmoid $(\frac{1}{1+e^{-t}})$, hyperbolic tangent $(\frac{e^t - e^{-t}}{e^t + e^{-t}})$, and softplus $(\frac{1}{c}\log(1+e^{ct})$ for some $c > 0)$ as a smooth approximation of ReLU; (c) $\Omega$ is the group Lasso.*

*Then the sequence $\{M_k\}$ converges to a stationary point of $L(M)$ under the conditions of Theorem 1.*

## B.1 SUFFICIENT DESCENT PROPERTY ALONG LYAPUNOV FUNCTION

Let $P_k := (M_k, \Gamma_k)$, and $Q_k := (P_k, g_{k-1}), k \in \mathbb{N}$. In the following, we present the sufficient descent property of $Q_k$ along the Lyapunov function $F$.

**Lemma.** Suppose that $\mathcal{L}$ is continuously differentiable and $\nabla\mathcal{L}$ is Lipschitz continuous with a constant $Lip > 0$, $C = \max|W_0|$ is the max value of the pretrained model parameters $W_0$. Let $\{Q_k\}$ be a sequence generated by SLBI with a finite initialization. If $0 < \alpha < \frac{2}{\kappa(Lip*C+\nu^{-1})}$, then

$$F(Q_{k+1}) \le F(Q_k) - \rho\|Q_{k+1} - Q_k\|_2^2,$$

where $\rho := \frac{1}{\kappa} - \frac{\alpha(Lip*C+\nu^{-1})}{2}$.

*Proof.* By the optimality condition of (20a) and also the inclusion $p_k = [0, g_k]^T \in \partial R(P_k)$, there holds

$$\kappa(\alpha\nabla\bar{\mathcal{L}}(P_k) + p_{k+1} - p_k) + P_{k+1} - P_k = 0,$$

which implies

$$-\langle\alpha\nabla\bar{\mathcal{L}}(P_k), P_{k+1} - P_k\rangle = \kappa^{-1}\|P_{k+1} - P_k\|_2^2 + D(\Gamma_{k+1}, \Gamma_k) \tag{23}$$

where

$$D(\Gamma_{k+1}, \Gamma_k) := \langle g_{k+1} - g_k, \Gamma_{k+1} - \Gamma_k\rangle.$$

Let $W_0 \odot M = \hat{W}$, with $\hat{\mathcal{L}}(M) = \mathcal{L}(W_0 \odot M)$,

$$\nabla\hat{\mathcal{L}}(M) = \sum\nabla\mathcal{L}(\hat{W}) * W_0 \tag{24}$$

Noting that $\bar{\mathcal{L}}(P) = \hat{\mathcal{L}}(M) + \frac{1}{2\nu}\|M - \Gamma\|_2^2 = \mathcal{L}(W_0 \odot M) + \frac{1}{2\nu}\|M - \Gamma\|_2^2$, and by the Lipschitz continuity of $\nabla\mathcal{L}(W)$ with constants $Lip > 0, C = \max|W_0| > 0$ implies $\nabla\bar{\mathcal{L}}$ is Lipschitz continuous with a constant $Lip * C + \nu^{-1}$. This implies

$$\bar{\mathcal{L}}(P_{k+1}) \leq \bar{\mathcal{L}}(P_k) + \langle\nabla\bar{\mathcal{L}}(P_k), P_{k+1} - P_k\rangle + \frac{Lip * C + \nu^{-1}}{2}\|P_{k+1} - P_k\|_2^2.$$

Substituting the above inequality into (23) yields

$$\alpha\bar{\mathcal{L}}(P_{k+1}) + D(\Gamma_{k+1}, \Gamma_k) + \rho\|P_{k+1} - P_k\|_2^2 \leq \alpha\bar{\mathcal{L}}(P_k). \tag{25}$$

Adding some terms in both sides of the above inequality and after some reformulations implies

$$\alpha\bar{\mathcal{L}}(P_{k+1}) + B_\Omega^{g_k}(\Gamma_{k+1}, \Gamma_k) \tag{26}$$
$$\leq \alpha\bar{\mathcal{L}}(P_k) + B_\Omega^{g_{k-1}}(\Gamma_k, \Gamma_{k-1}) - \rho\|P_{k+1} - P_k\|_2^2 - \left(D(\Gamma_{k+1}, \Gamma_k) + B_\Omega^{g_{k-1}}(\Gamma_k, \Gamma_{k-1}) - B_\Omega^{g_k}(\Gamma_{k+1}, \Gamma_k)\right)$$
$$= \alpha\bar{\mathcal{L}}(P_k) + B_\Omega^{g_{k-1}}(\Gamma_k, \Gamma_{k-1}) - \rho\|P_{k+1} - P_k\|_2^2 - B_\Omega^{g_{k+1}}(\Gamma_k, \Gamma_{k-1}) - B_\Omega^{g_{k-1}}(\Gamma_k, \Gamma_{k-1}),$$

where the final equality holds for $D(\Gamma_{k+1}, \Gamma_k) - B_\Omega^{g_k}(\Gamma_{k+1}, \Gamma_k) = B_\Omega^{g_{k+1}}(\Gamma_k, \Gamma_{k-1})$. That is,

$$F(Q_{k+1}) \leq F(Q_k) - \rho\|P_{k+1} - P_k\|_2^2 - B_\Omega^{g_{k+1}}(\Gamma_k, \Gamma_{k-1}) - B_\Omega^{g_{k-1}}(\Gamma_k, \Gamma_{k-1}) \tag{27}$$
$$\leq F(Q_k) - \rho\|P_{k+1} - P_k\|_2^2, \tag{28}$$

where the final inequality holds for $B_\Omega^{g_{k+1}}(\Gamma_k, \Gamma_{k-1}) \geq 0$ and $B_\Omega^{g_{k-1}}(\Gamma_k, \Gamma_{k-1}) \geq 0$. Thus, we finish the proof of this lemma. $\square$

Based on Lemma B.1, we directly obtain the following lemma.

**Lemma 1.** *Suppose that assumptions of Lemma B.1 hold. Suppose further that Assumption 1 (b)-(d) hold. Then*

*(i) both $\alpha\{\bar{\mathcal{L}}(P_k)\}$ and $\{F(Q_k)\}$ converge to the same finite value, and $\lim_{k\to\infty}B_\Omega^{g_k}(\Gamma_{k+1}, \Gamma_k) = 0$.*

*(ii) the sequence $\{(M_k, \Gamma_k, g_k)\}$ is bounded,*

*(iii) $\lim_{k\to\infty}\|P_{k+1} - P_k\|_2^2 = 0$ and $\lim_{k\to\infty}D(\Gamma_{k+1}, \Gamma_k) = 0$,*

*(iv) $\frac{1}{K}\sum_{k=0}^{K}\|P_{k+1} - P_k\|_2^2 \to 0$ at a rate of $\mathcal{O}(1/K)$.*

*Proof.* By (25), $\bar{\mathcal{L}}(P_k)$ is monotonically decreasing due to $D(\Gamma_{k+1}, \Gamma_k) \geq 0$. Similarly, by (28), $F(Q^k)$ is also monotonically decreasing. By the lower boundedness assumption of $\mathcal{L}(W)$, both $\bar{\mathcal{L}}(P)$ and $F(Q)$ are lower bounded by their definitions, i.e., (9) and (15), respectively. Therefore, both $\{\bar{\mathcal{L}}(P_k)\}$ and $\{F(Q_k)\}$ converge, and it is obvious that $\lim_{k\to\infty}F(Q_k) \geq \lim_{k\to\infty}\alpha\bar{\mathcal{L}}(P_k)$. By (27),

$$B_\Omega^{g_{k-1}}(\Gamma_k, \Gamma_{k-1}) \leq F(Q_k) - F(Q_{k+1}), \ k = 1, \ldots.$$

By the convergence of $F(Q_k)$ and the nonegativeness of $B_\Omega^{g_{k-1}}(\Gamma_k, \Gamma_{k-1})$, there holds

$$\lim_{k \to \infty} B_\Omega^{g_{k-1}}(\Gamma_k, \Gamma_{k-1}) = 0.$$

By the definition of $F(Q_k) = \alpha\bar{\mathcal{L}}(P_k) + B_\Omega^{g_{k-1}}(\Gamma_k, \Gamma_{k-1})$ and the above equality, it yields

$$\lim_{k \to \infty} F(Q_k) = \lim_{k \to \infty} \alpha\bar{\mathcal{L}}(P_k).$$

Since $L(M)$ has bounded level sets, then $M_k$ is bounded. By the definition of $\bar{\mathcal{L}}(M, \Gamma)$ and the finiteness of $\bar{\mathcal{L}}(M_k, \Gamma_k)$, $\Gamma_k$ is also bounded due to $M_k$ is bounded. The boundedness of $g_k$ is due to $g_k \in \partial\Omega(\Gamma_k)$, condition (d), and the boundedness of $\Gamma_k$.

By (28), summing up (28) over $k = 0, 1, \ldots, K$ yields

$$\sum_{k=0}^{K} \left( \rho\|P_{k+1} - P_k\|^2 + D(\Gamma_{k+1}, \Gamma_k) \right) < \alpha\bar{\mathcal{L}}(P_0) < \infty. \tag{29}$$

Letting $K \to \infty$ and noting that both $\|P_{k+1} - P_k\|^2$ and $D(\Gamma_{k+1}, \Gamma_k)$ are nonnegative, thus

$$\lim_{k \to \infty} \|P_{k+1} - P_k\|^2 = 0, \quad \lim_{k \to \infty} D(\Gamma_{k+1}, \Gamma_k) = 0.$$

Again by (29),

$$\frac{1}{K} \sum_{k=0}^{K} \left( \rho\|P_{k+1} - P_k\|^2 + D(\Gamma_{k+1}, \Gamma_k) \right) < K^{-1}\alpha\bar{\mathcal{L}}(P_0),$$

which implies $\frac{1}{K} \sum_{k=0}^{K} \|P_{k+1} - P_k\|^2 \to 0$ at a rate of $\mathcal{O}(1/K)$. $\qquad\square$

## B.2 RELATIVE ERROR PROPERTY

In this subsection, we provide the bound of subgradient by the discrepancy of two successive iterates. By the definition of $F$ (15),

$$H_{k+1} := \begin{pmatrix} \alpha\nabla_M\bar{\mathcal{L}}(M_{k+1}, \Gamma_{k+1}) \\ \alpha\nabla_\Gamma\bar{\mathcal{L}}(M_{k+1}, \Gamma_{k+1}) + g_{k+1} - g_k \\ \Gamma_k - \Gamma_{k+1} \end{pmatrix} \in \partial F(Q_{k+1}), \ k \in \mathbb{N}. \tag{30}$$

**Lemma.** Under assumptions of Lemma 1, then

$$\|H_{k+1}\| \le \rho_1\|Q_{k+1} - Q_k\|, \text{ for } H_{k+1} \in \partial F(Q_{k+1}), \ k \in \mathbb{N},$$

where $\rho_1 := 2\kappa^{-1} + 1 + \alpha(Lip * C + 2\nu^{-1})$. Moreover, $\frac{1}{K} \sum_{k=1}^{K} \|H_k\|^2 \to 0$ at a rate of $\mathcal{O}(1/K)$.

*Proof.* Note that

$$\nabla_M\bar{\mathcal{L}}(M_{k+1}, \Gamma_{k+1}) = (\nabla_M\bar{\mathcal{L}}(M_{k+1}, \Gamma_{k+1}) - \nabla_M\bar{\mathcal{L}}(M_{k+1}, \Gamma_k)) \tag{31}$$
$$+ (\nabla_M\bar{\mathcal{L}}(M_{k+1}, \Gamma_k) - \nabla_M\bar{\mathcal{L}}(M_k, \Gamma_k)) + \nabla_M\bar{\mathcal{L}}(M_k, \Gamma_k).$$

By the definition of $\bar{\mathcal{L}}$ (see (9)),

$$\|\nabla_M\bar{\mathcal{L}}(M_{k+1}, \Gamma_{k+1}) - \nabla_M\bar{\mathcal{L}}(M_{k+1}, \Gamma_k)\| = \nu^{-1}\|\Gamma_k - \Gamma_{k+1}\|,$$
$$\|\nabla_M\bar{\mathcal{L}}(M_{k+1}, \Gamma_k) - \nabla_M\bar{\mathcal{L}}(M_k, \Gamma_k)\| = \|(\nabla\mathcal{L}(M_{k+1}) - \nabla\mathcal{L}(M_k)) + \nu^{-1}(M_{k+1} - M_k)\|$$
$$\le (Lip * C + \nu^{-1})\|M_{k+1} - M_k\|,$$

where the last inequality holds for the Lipschitz continuity of $\nabla\mathcal{L}$ with a constant $Lip > 0$, and $C = \max|W_0|$. By (21a),

$$\|\nabla_M\bar{\mathcal{L}}(M_k, \Gamma_k)\| = (\kappa\alpha)^{-1}\|M_{k+1} - M_k\|.$$

Substituting the above (in)equalities into (31) yields

$$\|\nabla_M\bar{\mathcal{L}}(M_{k+1}, \Gamma_{k+1})\| \le \left[(\kappa\alpha)^{-1} + Lip * C + \nu^{-1}\right] \cdot \|M_{k+1} - M_k\| + \nu^{-1}\|\Gamma_{k+1} - \Gamma_k\|$$

Thus,

$$\|\alpha \nabla_M \bar{\mathcal{L}}(M_{k+1}, \Gamma_{k+1})\| \leq \left[\kappa^{-1} + \alpha(Lip * C + \nu^{-1})\right] \cdot \|M_{k+1} - M_k\| + \alpha\nu^{-1}\|\Gamma_{k+1} - \Gamma_k\|. \tag{32}$$

By (21c), it yields

$$g_{k+1} - g_k = \kappa^{-1}(\Gamma_k - \Gamma_{k+1}) - \alpha\nabla_\Gamma \bar{\mathcal{L}}(M_k, \Gamma_k).$$

Noting that $\nabla_\Gamma \bar{\mathcal{L}}(M_k, \Gamma_k) = \nu^{-1}(\Gamma_k - M_k)$, and after some simplifications yields

$$\begin{aligned}
\|\alpha\nabla_\Gamma \bar{\mathcal{L}}(M_{k+1}, \Gamma_{k+1}) + g_{k+1} - g_k\| &= \|(\kappa^{-1} - \alpha\nu^{-1}) \cdot (\Gamma_k - \Gamma_{k+1}) + \alpha\nu^{-1}(M_k - M_{k+1})\| \\
&\leq \alpha\nu^{-1}\|M_k - M_{k+1}\| + (\kappa^{-1} - \alpha\nu^{-1})\|\Gamma_k - \Gamma_{k+1}\|,
\end{aligned} \tag{33}$$

where the last inequality holds for the triangle inequality and $\kappa^{-1} > \alpha\nu^{-1}$ by the assumption.

By (32), (33), and the definition of $H_{k+1}$ (30), there holds

$$\begin{aligned}
\|H_{k+1}\| &\leq \left[\kappa^{-1} + \alpha(Lip * C + 2\nu^{-1})\right] \cdot \|M_{k+1} - M_k\| + (\kappa^{-1} + 1)\|\Gamma_{k+1} - \Gamma_k\| \\
&\leq \left[2\kappa^{-1} + 1 + \alpha(Lip * C + 2\nu^{-1})\right] \cdot \|P_{k+1} - P_k\| \\
&\leq \left[2\kappa^{-1} + 1 + \alpha(Lip * C + 2\nu^{-1})\right] \cdot \|Q_{k+1} - Q_k\|.
\end{aligned} \tag{34}$$

By (34) and Lemma 1(iv), $\frac{1}{K}\sum_{k=1}^{K} \|H_k\|^2 \to 0$ at a rate of $\mathcal{O}(1/K)$.

This finishes the proof of this lemma. $\qquad\square$

### B.3 KURDYKA-ŁOJASIEWICZ PROPERTY

To introduce the definition of the Kurdyka-Łojasiewicz (KL) property, we need some notions and notations from variational analysis, which can be found in Rockafellar & Wets (1998).

The notion of subdifferential plays a central role in the following definitions. For each $\mathbf{x} \in \mathrm{dom}(h) := \{\mathbf{x} \in \mathbb{R}^p : h(\mathbf{x}) < +\infty\}$, the *Fréchet subdifferential* of $h$ at $\mathbf{x}$, written $\widehat{\partial}h(\mathbf{x})$, is the set of vectors $\mathbf{v} \in \mathbb{R}^p$ which satisfy

$$\lim_{\mathbf{y}\neq\mathbf{x}, \mathbf{y}\to\mathbf{x}} \inf \frac{h(\mathbf{y}) - h(\mathbf{x}) - \langle\mathbf{v}, \mathbf{y} - \mathbf{x}\rangle}{\|\mathbf{x} - \mathbf{y}\|} \geq 0.$$

When $\mathbf{x} \notin \mathrm{dom}(h)$, we set $\widehat{\partial}h(\mathbf{x}) = \varnothing$. The *limiting-subdifferential* (or simply *subdifferential*) of $h$ introduced in Mordukhovich (2006), written $\partial h(\mathbf{x})$ at $\mathbf{x} \in \mathrm{dom}(h)$, is defined by

$$\partial h(\mathbf{x}) := \{\mathbf{v} \in \mathbb{R}^p : \exists \mathbf{x}^k \to \mathbf{x}, \ h(\mathbf{x}^k) \to h(\mathbf{x}), \ \mathbf{v}^k \in \widehat{\partial}h(\mathbf{x}^k) \to \mathbf{v}\}. \tag{35}$$

A necessary (but not sufficient) condition for $\mathbf{x} \in \mathbb{R}^p$ to be a minimizer of $h$ is $\mathbf{0} \in \partial h(\mathbf{x})$. A point that satisfies this inclusion is called *limiting-critical* or simply *critical*. The distance between a point $\mathbf{x}$ to a subset $\mathcal{S}$ of $\mathbb{R}^p$, written $\mathrm{dist}(\mathbf{x}, \mathcal{S})$, is defined by $\mathrm{dist}(\mathbf{x}, \mathcal{S}) = \inf\{\|\mathbf{x} - \mathbf{s}\| : \mathbf{s} \in \mathcal{S}\}$, where $\|\cdot\|$ represents the Euclidean norm.

Let $h : \mathbb{R}^p \to \mathbb{R} \cup \{+\infty\}$ be an extended-real-valued function (respectively, $h : \mathbb{R}^p \rightrightarrows \mathbb{R}^q$ be a point-to-set mapping), its *graph* is defined by

$$\mathrm{Graph}(h) := \{(\mathbf{x}, y) \in \mathbb{R}^p \times \mathbb{R} : y = h(\mathbf{x})\},$$
$$(\text{resp. } \mathrm{Graph}(h) := \{(\mathbf{x}, \mathbf{y}) \in \mathbb{R}^p \times \mathbb{R}^q : \mathbf{y} \in h(\mathbf{x})\}),$$

and its domain by $\mathrm{dom}(h) := \{\mathbf{x} \in \mathbb{R}^p : h(\mathbf{x}) < +\infty\}$ (resp. $\mathrm{dom}(h) := \{\mathbf{x} \in \mathbb{R}^p : h(\mathbf{x}) \neq \varnothing\}$). When $h$ is a proper function, i.e., when $\mathrm{dom}(h) \neq \varnothing$, the set of its global minimizers (possibly empty) is denoted by

$$\arg\min h := \{\mathbf{x} \in \mathbb{R}^p : h(\mathbf{x}) = \inf h\}.$$

The KL property (Łojasiewicz, 1963; 1993; Kurdyka, 1998; Bolte et al., 2007a;b) plays a central role in the convergence analysis of nonconvex algorithms (Attouch et al., 2013; Wang et al., 2019). The following definition is adopted from Bolte et al. (2007b).

**Definition 1.** *[Kurdyka-Łojasiewicz property] A function $h$ is said to have the Kurdyka-Łojasiewicz (KL) property at $\bar{u} \in \text{dom}(\partial h) := \{v \in \mathbb{R}^n | \partial h(v) \neq \emptyset\}$, if there exists a constant $\eta \in (0, \infty)$, a neighborhood $\mathcal{N}$ of $\bar{u}$ and a function $\phi : [0, \eta) \to \mathbb{R}_+$, which is a concave function that is continuous at $0$ and satisfies $\phi(0) = 0$, $\phi \in \mathcal{C}^1((0, \eta))$, i.e., $\phi$ is continuous differentiable on $(0, \eta)$, and $\phi'(s) > 0$ for all $s \in (0, \eta)$, such that for all $u \in \mathcal{N} \cap \{u \in \mathbb{R}^n | h(\bar{u}) < h(u) < h(\bar{u}) + \eta\}$, the following inequality holds*

$$\phi'(h(u) - h(\bar{u})) \cdot \text{dist}(0, \partial h(u)) \geq 1. \tag{36}$$

*If $h$ satisfies the KL property at each point of $\text{dom}(\partial h)$, $h$ is called a KL function.*

KL functions include semialgebraic functions, real analytic functions, continuous subanalytic functions (Bolte et al., 2007a) and locally strongly convex functions, tame functions defined in some o-minimal structures (Kurdyka, 1998; Bolte et al., 2007b). In the following, we provide some important examples that satisfy the Kurdyka-Łojasiewicz property.

**Definition 2.** *[Semialgebraic]*

(a) *A function $h : \mathbb{R}^p \to \mathbb{R} \cup \{+\infty\}$ (resp. a point-to-set mapping $h : \mathbb{R}^p \rightrightarrows \mathbb{R}^q$) is called semialgebraic if its graph $\text{Graph}(h)$ is a semialgebraic set.*

(b) *A set $\mathcal{D} \subset \mathbb{R}^p$ is called semialgebraic (Bochnak et al., 1998) if it can be represented as*

$$\mathcal{D} = \bigcup_{i=1}^{s} \bigcap_{j=1}^{t} \{\mathbf{x} \in \mathbb{R}^p : P_{ij}(\mathbf{x}) = 0, Q_{ij}(\mathbf{x}) > 0\},$$

*where $P_{ij}, Q_{ij}$ are real polynomial functions for $1 \leq i \leq s, 1 \leq j \leq t$.*

According to (Łojasiewicz, 1965; Bochnak et al., 1998) and (Shiota, 1997, I.2.9, page 52), the class of semialgebraic sets are stable under the operation of finite union, finite intersection, Cartesian product or complementation. Some typical examples include polynomial functions, the indicator function of a semialgebraic set, and the Euclidean norm (Bochnak et al., 1998, page 26).

**Definition 3.** *[Real analytic] A function $h$ with domain an open set $U \subset \mathbb{R}$ and range the set of either all real or complex numbers, is said to be **real analytic** at $u$ if the function $h$ may be represented by a convergent power series on some interval of positive radius centered at $u$: $h(x) = \sum_{j=0}^{\infty} \alpha_j(x-u)^j$, for some $\{\alpha_j\} \subset \mathbb{R}$. The function is said to be **real analytic** on $V \subset U$ if it is real analytic at each $u \in V$ (Krantz & Parks, 2002, Definition 1.1.5). The real analytic function $f$ over $\mathbb{R}^p$ for some positive integer $p > 1$ can be defined similarly.*

*According to Krantz & Parks (2002), typical real analytic functions include polynomials, exponential functions, and the logarithm, trigonometric and power functions on any open set of their domains. One can verify whether a multivariable real function $h(\mathbf{x})$ on $\mathbb{R}^p$ is analytic by checking the analyticity of $g(t) := h(\mathbf{x} + t\mathbf{y})$ for any $\mathbf{x}, \mathbf{y} \in \mathbb{R}^p$.*

### B.4 KL PROPERTY IN DEEP LEARNING AND PROOF OF COROLLARY 1

In the following, we consider the deep neural network training problem. Consider a $l$-layer feedforward neural network including $l - 1$ hidden layers of the neural network. Particularly, let $d_i$ be the number of hidden units in the $i$-th hidden layer for $i = 1, \ldots, l-1$.

Let $d_0$ and $d_l$ be the number of units of input and output layers, respectively. Let $W^i \in R^{d_i \times d_{i-1}}$ be the weight matrix between the $(i-1)$-th layer and the $i$-th layer for any $i = 1, \ldots l$[1].

According to Theorem 2, one major condition is to verify the introduced Lyapunov function $F$ defined in (15) satisfies the Kurdyka-Łojasiewicz property. For this purpose, we need an extension of semialgebraic set, called the *o-minimal structure* (see, for instance Coste (1999), van den Dries (1986), Kurdyka (1998), Bolte et al. (2007b)). The following definition is from Bolte et al. (2007b).

**Definition 4.** *[o-minimal structure] An o-minimal structure on $(\mathbb{R}, +, \cdot)$ is a sequence of boolean algebras $\mathcal{O}_n$ of "definable" subsets of $\mathbb{R}^n$, such that for each $n \in \mathbb{N}$*

---

[1]To simplify notations, we regard the input and output layers as the 0-th and the $l$-th layers, respectively, and absorb the bias of each layer into $W^i$.

    (i) *the elements of $\mathcal{O}_1$ are exactly finite unions of intervals and points.*

    (ii) *$\mathcal{O}_n$ contains the family of algebraic subsets of $\mathbb{R}^n$, that is, every set of the form*

    (iii) *if $A$ belongs to $\mathcal{O}_n$, then $A \times \mathbb{R}$ and $\mathbb{R} \times A$ belong to $\mathcal{O}_{n+1}$;*

    (iv) *if $\Pi : \mathbb{R}^{n+1} \to \mathbb{R}^n$ is the canonical projection onto $\mathbb{R}^n$, then for any $A$ in $\mathcal{O}_{n+1}$, the set $\Pi(A)$ belongs to $\mathcal{O}_n$;*

$$\{x \in \mathbb{R}^n : p(x) = 0\},$$

*where $p : \mathbb{R}^n \to \mathbb{R}$ is a polynomial function.*

Based on the definition of o-minimal structure, we can show the definition of the *definable function*.

**Definition 5.** *[Definable function] Given an o-minimal structure $\mathcal{O}$ (over $(\mathbb{R}, +, \cdot)$), a function $f : \mathbb{R}^n \to \mathbb{R}$ is said to be definable in $\mathcal{O}$ if its graph belongs to $\mathcal{O}_{n+1}$.*

According to van den Dries & Miller (1996); Bolte et al. (2007b), there are some important facts of the o-minimal structure, shown as follows.

    (i) The o-minimal structure is stable under the sum, composition, the inf-convolution and several other classical operations of analysis.

    (iI) The collection of *semialgebraic* sets is an o-minimal structure. Recall the semialgebraic sets are Bollean combinations of sets of the form

$$\{x \in \mathbb{R}^n : p(x) = 0, q_1(x) < 0, \ldots, q_m(x) < 0\},$$

where $p$ and $q_i$'s are polynomial functions in $\mathbb{R}^n$.

    (iiI) There exists an o-minimal structure that contains the sets of the form

$$\{(x, t) \in [-1, 1]^n \times \mathbb{R} : f(x) = t\}$$

where $f$ is real-analytic around $[-1, 1]^n$.

    (iV) There exists an o-minimal structure that contains simultaneously the graph of the exponential function $\mathbb{R} \ni x \mapsto \exp(x)$ and all semialgebraic sets.

The Kurdyka-Łojasiewicz property for the smooth definable function and non-smooth definable function were established in (Kurdyka, 1998, Theorem 1) and (Bolte et al., 2007b, Theorem 14), respectively. Now we are ready to present the proof of Corollary 1.

*Proof.* [Proof of Corollary 1] To justify this corollary, we only need to verify the associated Lyapunov function $F$ satisfies Kurdyka-Łojasiewicz inequality. In this case and by (22), $F$ can be rewritten as follows

$$F(\mathcal{M}, \Gamma, \mathcal{G}) = \alpha \left( \mathcal{L}(M, \Gamma) + \frac{1}{2\nu} \|M - \Gamma\|^2 \right) + \Omega(\Gamma) + \Omega^*(g) - \langle \Gamma, g \rangle.$$

Because $\mathcal{L}$ and $\sigma_i$'s are definable by assumptions, then $\mathcal{L}(M, \Gamma)$ are definable as compositions of definable functions.

Moreover, according to Krantz & Parks (2002), $\|M - \Gamma\|^2$ and $\langle \Gamma, g \rangle$ are semi-algebraic and thus definable. Since the group Lasso $\Omega(\Gamma) = \sum_g \|\Gamma\|_2$ is the composition of $l_2$ and $l_1$ norms, and the conjugate of group Lasso penalty is the maximum of group $l_2$-norm, *i.e.* $\Omega^*(\Gamma) = \max_g \|\Gamma_g\|_2$, where the $l_2$, $l_1$, and $l_\infty$ norms are definable, hence the group Lasso and its conjugate are definable as compositions of definable functions. Therefore, $F$ is definable and hence satisfies Kurdyka-Łojasiewicz inequality by (Kurdyka, 1998, Theorem 1).

The verifications of other cases listed in assumptions can be found in the proof of (Zeng et al., 2019, Proposition 1). This finishes the proof of this corollary. $\square$

### B.5  PROOF OF THEOREM 2

Our analysis is mainly motivated by a paper (Benning et al., 2017), as well as the influential work (Attouch et al., 2013). According to Lemma 2.6 in Attouch et al. (2013), there are mainly four ingredients in the analysis, that is, the *sufficient descent property*, *relative error property*, *continuity property* of the generated sequence and the *Kurdyka-Łojasiewicz property* of the function. More specifically, we first establish the *sufficient descent property* of the generated sequence via exploiting the Lyapunov function $F$ (see, (15)) in Lemma B.1 in Section B.1, and then show the *relative error property* of the sequence in Lemma B.2 in Section B.2. The *continuity property* is guaranteed by the continuity of $\bar{\mathcal{L}}(M, \Gamma)$ and the relation $\lim_{k \to \infty} B_{\Omega}^{g_k}(\Gamma_{k+1}, \Gamma_k) = 0$ established in Lemma 1(i) in Section B.1. Thus, together with the Kurdyka-Łojasiewicz assumption of $F$, we establish the global convergence of SLBI following by (Attouch et al., 2013, Lemma 2.6).

Let $(\bar{W}, \bar{\Gamma}, \bar{g})$ be a critical point of $F$, then the following holds

$$
\begin{aligned}
\partial_M F(\bar{M}, \bar{\Gamma}, \bar{g}) &= \alpha(\nabla \mathcal{L}(\bar{M}) + \nu^{-1}(\bar{M} - \bar{\Gamma})) = 0, \\
\partial_\Gamma F(\bar{M}, \bar{\Gamma}, \bar{g}) &= \alpha \nu^{-1}(\bar{\Gamma} - \bar{M}) + \partial \Omega(\bar{\Gamma}) - \bar{g} \ni 0, \\
\partial_g F(\bar{M}, \bar{\Gamma}, \bar{g}) &= \bar{\Gamma} - \partial \Omega^*(\bar{g}) \ni 0.
\end{aligned}
\tag{37}
$$

By the final inclusion and the convexity of $\Omega$, it implies $\bar{g} \in \partial \Omega(\bar{\Gamma})$. Plugging this inclusion into the second inclusion yields $\alpha \nu^{-1}(\bar{\Gamma} - \bar{M}) = 0$. Together with the first equality imples

$$
\nabla \bar{\mathcal{L}}(\bar{M}, \bar{\Gamma}) = 0, \quad \nabla \mathcal{L}(\bar{M}) = 0.
$$

This finishes the proof of this theorem.

