# OpenReview forum: "Adaptive Pruning of Pretrained Transformer via Differential Inclusions"
_ICLR.cc/2025/Conference — ICLR 2025 Poster_

### Official Review · Reviewer_P1Rj · 2024-10-28

**Soundness:** 3
**Presentation:** 3
**Contribution:** 3
**Rating:** 6
**Confidence:** 3

**Summary:**

This paper presents a pruning method of pretrained transformers at any desired ratio within a single pruning stage, based on a differential inclusion for a mask parameter.

**Strengths:**

1.	Clear motivation.
2.	The theoretical proof and experiment are quite sufficient.

**Weaknesses:**

1.	The results of ablation studies are insufficient to demonstrate the effectiveness of the proposed method for the following reasons: 1) For DeiT-Small and Swin-Tiny models, the proposed SPP achieves higher accuracy with more parameters and higher or equal FLOPS, which does not indicate that SPP is superior. 2) Conducting experiments solely with the DessiLBI method lacks generalizability.
2.	Figure 1 and tables lack legends and comments.
3.	"FLOPs" is generally not written as "Flops" when used as an abbreviation for "floating point operations".

**Questions:**

1.	The manuscript does not describe how low-rank compression can be applied to paired modules of transformers.
2.	The reviewer is concerned about the actual running time of the proposed method and how much its efficiency improves compared to methods with a single pruning rate.
3.	There seem to be some errors in the details of the manuscript, for example, "Specifically, note that for each element i, V (i) ≤ 1 if V (i) = 0, and is equal to 1 when it becomes non-zeros".

---

> ### Author Response · Authors · 2024-11-15
>
> **Weakness 1: The results are insufficient.**
>
> As shown in Table 1, for Deit-Small, with the same number of flops 2.6, WDPruning has an accuracy of 78.4% compared to 78.9% for our method. For Deit-Base, we outperform all methods with 81.2% accuracy and keep fewer flops, only 6.9.
>
> As we explained to Reviewer 1jDf, our method not only provides better numerical results but also introduces a new pruning approach based on differential inclusion. We identify the sparse structure of the model in the inverse space and map it to the solution path. The most important parameters are learned first, followed by the less important ones, allowing us to prune based on each parameter's importance. Therefore, our approach learns a regularization path, eliminating the need to re-prune the network for each sparsity level.As shown in Figure 2, our method is very consistent, ensuring that adaptive pruning is done effectively.
>
> **Weakness 2: Figure 1 and tables lack legends and comments.**
>
> We will rewrite the comments for Figure 1 and Table 1 to make them clearer and easier to understand.
>
> **Weakness 3: Typo error.**
>
> Thank you for pointing out the typo . We will correct this and improve our writing when revisiting the manuscript.
>
> **Question 1: The application of low-rank compression to paired modules of transformers.**
>
> The detailed application equations can be found in Section 3.1.
>
> **Question 2: The actual running time of the proposed method.**
>
> The latency times for the uncompressed and compressed CLIP models are shown in Table 7. The search epochs and training epochs are listed in Table 6.

---

> ### Author Response · Authors · 2024-11-22
>
> Dear Reviewers,
>
> We have worked hard to address your comments in detail and would greatly appreciate any additional feedback or clarification. If anything remains unclear or if more information is needed, please let us know how we can assist. Your feedback is important to us, and we look forward to hearing from you whenever it is convenient.
>
> Best regards,

---

### Official Review · Reviewer_1jDf · 2024-11-01

**Soundness:** 2
**Presentation:** 3
**Contribution:** 2
**Rating:** 6
**Confidence:** 3

**Summary:**

The paper proposes SPP (Solution Path Pruning), a novel approach to mask-based pruning for pretrained transformers. SPP leverages differential inclusions to create a dynamic solution path that produces a range of sparsity levels in a single pruning stage, eliminating the need for multiple retraining steps for different compression ratios. The method applies fine-grained, pair-wise shared masks across transformer layers, including attention heads and MLP layers, achieving high flexibility and model performance retention. SPP is tested on various datasets and transformer backbones, showing efficiency improvements with minimal accuracy loss.

**Strengths:**

**Adaptive Solution Path for Pruning**
   - Unlike traditional mask-based pruning, SPP generates models with different sparsity levels in a single pruning run, allowing for a Transformer Weight Family adaptable to varying hardware or performance needs without retraining.

**Flexible, Fine-Grained Pruning Strategy**
   - SPP’s pair-wise shared mask strategy applies pruning at the smallest functional units within transformers (e.g., query-key and value-output pairs), allowing for greater flexibility and effectiveness in compression compared to conventional methods.

**Reduced Computational Cost**
   - The SPP method reduces the need for repeated pruning stages, resulting in significant cost savings, especially for large-scale transformer models.

**Weaknesses:**

**Marginal Improvement Over Existing Methods**
   - Although the method introduces adaptive pruning, it does not fundamentally change the mask-based pruning paradigm. The improvements, while novel in terms of execution, may appear incremental compared to existing mask-based and structural pruning strategies.

**Lack of Broad Comparison with Other Mask-Based Methods**
   - The paper does not provide an in-depth comparison with other advanced mask-based pruning techniques, making it difficult to fully assess SPP's advantages in performance and efficiency.

**Questions:**

- How does the method scale to very large transformers (e.g., GPT-3 scale)?
- What is the stability of the solution path across different random seeds?
- How sensitive is the method to the choice of hyperparameters κ and λ?
- Can this method be extended to dynamic/runtime pruning scenarios?

---

> ### Author Response · Authors · 2024-11-15
>
> **Weakness 1: Marginal Improvement Over Existing Methods**
>
> Traditional mask-based pruning methods, while effective in fixed-ratio compression, are not efficient in adaptive pruning, which is important for deployment in many scenarios. This paper provides a comprehensive solution. First, by adopting the perspective of differential inclusion, we can efficiently obtain a solution path from sparse to dense, making it well-suited for adaptive training. Moreover, our dynamics enjoys the inverse scale space property:  important features will select at earlier epochs, and while be consistently maintained in later epochs (figure 2). Finally, we establish the convergence results, ensuring the reliability of our algorithm.
>
> **Weakness 2: Lack of Broad Comparison with Other Mask-Based Methods**
>
> In Table 1, we have compared with several masked based pruning methods, such as UPop [1] and VitSlimm [2], which have achieved state-of-the-art results.
>
> [1]Shi, Dachuan, et al. "Upop: Unified and progressive pruning for compressing vision-language transformers." International Conference on Machine Learning. PMLR, 2023.
>
> [2]Chavan, Arnav, et al. "Vision transformer slimming: Multi-dimension searching in continuous optimization space." Proceedings of the IEEE/CVF Conference on Computer Vision and Pattern Recognition. 2022.
>
> **Question 1: How does the method scale to very large transformers？**
>
> The detailed algorithm is shown in Algorithm 2. As stated in line 762-765, we apply zero shot pruning to LLMs with our method. Since there is no weight update, our pruning method is efficient while preserving the generalization ability.
>
> **Question 2: What is the stability of the solution path across different random seeds?**
>
> Our method is robust across seeds.
>
> **Question 3: How sensitive is the method to the choice of hyperparameters κ and λ?**
>
> Our method is not sensitive to hyperparameters as is shown in Table 8. In fact, as long as κ and λ are set in a way that allows the inverse space to learn the important parameters, the method will work effectively.
>
> **Question 4: Can this method be extended to dynamic/runtime pruning scenarios?**
>
> In theory, we could add weight updates during algorithm iterations to achieve runtime pruning. However, this would result in additional training costs and make the model's loss function too complex, which could prevent proper convergence. Besides,  It would increase the training time. Starting from a fixed, pretrained checkpoint is more efficient.

---

> ### Author Response · Authors · 2024-11-22
>
> Dear Reviewers,
>
> We have worked hard to address your comments in detail and would greatly appreciate any additional feedback or clarification. If anything remains unclear or if more information is needed, please let us know how we can assist. Your feedback is important to us, and we look forward to hearing from you whenever it is convenient.
>
> Best regards,

---

### Official Review · Reviewer_MvCV · 2024-11-03

**Soundness:** 3
**Presentation:** 3
**Contribution:** 3
**Rating:** 6
**Confidence:** 3

**Summary:**

This paper proposes a method to prune pretrained transformers at any desired ratio within a single pruning stage. The proposed method enjoys a theoretical analysis to guarantee the global convergence. Extensive experiments validate the effectiveness of the proposed method.

**Strengths:**

1. This paper is motivated well. The method for any desired pruning ratio is needed in many real-world applications. And the proposed method is able to achieve this.
2. This paper identifies the limitation of Lasso and develops a differential inclusion-based method to achieve various compression ratio pruning.
3. There is a sound theoretical analysis to guarantee the global convergence of the method. A detailed proof is included in the appendix.
4. Experimental results are strong. Many experiments are conducted, including ViTs for image classification, CLIP, and even large language models.

**Weaknesses:**

1. Although the authors claim the proposed method significantly reducing the cost of model pruning. The training cost is not reported in this paper. It is better to introduce how long the search stage is. And make a comparison for training cost between different methods.
2. The ablation studies are weak. Experiments demonstrate the strong performance of the proposed SPP. But it is hard for the reader to figure out why the proposed method is effective. More ablation studies are needed. For example, make comparison for weight-based pruning and mask-based pruning. Why does the weight-based pruning not be applicable to the Transformer? What if using Lasso for the searching stage. By optimizating Eq. 7, Lasso is able to achieve different level of sparsity during training.

**Questions:**

1. For pruning large language models, why to combine the propsed SPP with the RIA pruning metric? However, pruning CV models (ViTs and CLIP) does not use pruning metric. What is the difference between pruning CV models and LLMs?
2. In line 207, it seems that the dimension is d_1 rather than d.

---

> ### Author Response · Authors · 2024-11-15
>
> **Weakness 1: Training cost of the proposed method.**
>
> The training cost of our proposed method is provided in detail in Table 6 and Table 8 of the Appendix. We have listed both the search cost and the fine-tuning cost. Our fine-tuning cost for the Deit models is 300 epochs, which is the same as other methods such as WDpruning[1] and UPop[2].
> | Method       | Search Epochs | Finetuning Epochs  |
> |-------------|-----------|-------------|
> | UPop   | 60            | 300               |
> | WDpruning | 100           | 300          |
> | ViT-Slim[3]   | 50             | 300       |
> | Ours  | 40             | 300              |
>
> [1]Yu, Fang, et al. "Width & depth pruning for vision transformers." Proceedings of the AAAI Conference on Artificial Intelligence. Vol. 36. No. 3. 2022.
>
> [2]Shi, Dachuan, et al. "Upop: Unified and progressive pruning for compressing vision-language transformers." International Conference on Machine Learning. PMLR, 2023.
>
> [3]Chavan, Arnav, et al. "Vision transformer slimming: Multi-dimension searching in continuous optimization space." Proceedings of the IEEE/CVF Conference on Computer Vision and Pattern Recognition. 2022.
>
> **Weakness 2: Comparison between weight-based pruning and mask-based pruning.**
>
> Weight-based pruning has already been shown to be less effective compared to mask-based pruning in the VitSlimm[1] method. DessiLBI[2] method has the same pruning metric with weight-based pruning. The comparison with DessiLBI method can be tought to be comparison with weight-based pruning. While Lasso can be used for mask-based pruning (line 246, Eq.7), it is computationally expensive to adaptively train nonlinear models which do not have a closed-form solution path, as we have for linear models (line 249-250). That means, we need to optimize the loss in Eq. (7) for each regularization parameter \lambda. Moreover, UPop and VitSlimm are advanced mask-based pruning methods that existed before our method. We compared these methods with ours in Table 1.
>
> [1]Chavan, Arnav, et al. "Vision transformer slimming: Multi-dimension searching in continuous optimization space." Proceedings of the IEEE/CVF Conference on Computer Vision and Pattern Recognition. 2022.
>
> [2]Fu, Yanwei, et al. "Dessilbi: Exploring structural sparsity of deep networks via differential inclusion paths." International Conference on Machine Learning. PMLR, 2020.
>
> **Question 1: How our method extends to pruning large language models.**
>
> The detailed algorithm is shown in Algorithm 2. For large language models, the fine-tuning cost is higher than for models like ViTs and CLIP. As mentioned in line 762, it is important for LLMs to maintain their generalization performance even after pruning. Current pruning methods for ViTs cannot preserve generalization. Therefore, we extended our method to zero-shot pruning with no weight update, where the RIA [4] method performs best.
>
> [4] Zhang, Yingtao, et al. "Plug-and-play: An efficient post-training pruning method for large language models." The Twelfth International Conference on Learning Representations. 2024.
>
> **Question 2: Typo error.**
>
> Thank you for pointing out the typos. We will make sure to improve our writing when revisiting the manuscript.

---

> ### Author Response · Authors · 2024-11-22
>
> Dear Reviewers,
>
> We have worked hard to address your comments in detail and would greatly appreciate any additional feedback or clarification. If anything remains unclear or if more information is needed, please let us know how we can assist. Your feedback is important to us, and we look forward to hearing from you whenever it is convenient.
>
> Best regards,

---

### Author Response · Authors · 2024-11-21

Dear ACs and Reviewers,

I hope this message finds you well. I apologize for any inconvenience, but I wanted to follow up on the rebuttals we submitted this week. We have put in our best effort to address the reviewers' concerns and are keen to clarify any remaining issues or misunderstandings. If possible, could the reviewers please provide your feedback on our rebuttals at your earliest convenience? We would be truly grateful if you could take the time to review our responses. We are more than willing to continue the discussion regarding our work.

Lastly, we thank all the reviewers and ACs for their time and efforts, regardless of the outcome of this paper.

Best regards,

---

### Comment · Area_Chair_FVCm · 2024-11-23
**Follow-Up Discussion on Author Feedback**

Dear PC Members,

Thank you for your valuable comments during the review period, which raised many interesting and insightful questions. The authors have now posted their feedback, and I encourage you to review their responses and engage in further discussion if necessary.

I understand that you may have a busy schedule, but your additional input is highly appreciated. Your contributions are crucial in ensuring a fair and well-rounded decision-making process.

Thank you once again for your continued support and dedication to ICLR.

Best regards,

AC

---

### Comment · Area_Chair_FVCm · 2024-12-01
**Action is required: Public Discussion Phase Ending Soon**

Dear PC memebers,

Thank you for your valuable comments during the review period, which raised many interesting and insightful questions.

Now the discussion period is coming to a close (less than one day for posing discussion with the authors)

PLEASE take a moment to review the authors’ responses if you haven’t done so already. Even if you decide not to update your evaluation, kindly confirm that you have reviewed the responses and that they do not change your assessment.

Best Regards,
AC

---

### Meta-Review · Area_Chair_FVCm · 2024-12-20

**Metareview:**

Pruning transformers is a crucial topic, and the proposed method, which utilizes differential inclusion for a mask parameter, appears reasonable and promising. The experimental results are significant, and all three reviewers provided positive scores. In my opinion, the questions raised were adequately addressed by the authors.

Unfortunately, the reviewers for this paper were unusually unresponsive and did not provide any additional feedback, despite several reminders. This may be because their initial comments were consistent and clear. Given the alignment of reviewer opinions, the recommendation for acceptance is straightforward and well-justified.

**Additional Comments On Reviewer Discussion:**

During the initial stage, the reviewers provided insightful comments and raised several important questions. However, none of the three reviewers offered any additional feedback in the subsequent stages, despite multiple reminders.

---

### Decision · Program_Chairs · 2025-01-22

Accept (Poster)